# The mechanism of replication stalling and recovery within repetitive DNA

Corella S. Casas-Delucchi[1,2], Manuel Daza-Martin[1,2], Sophie L. Williams[1] & Gideon Coster [1✉]

Accurate chromosomal DNA replication is essential to maintain genomic stability. Genetic evidence suggests that certain repetitive sequences impair replication, yet the underlying mechanism is poorly defined. Replication could be directly inhibited by the DNA template or indirectly, for example by DNA-bound proteins. Here, we reconstitute replication of mono-, di- and trinucleotide repeats in vitro using eukaryotic replisomes assembled from purified proteins. We find that structure-prone repeats are sufficient to impair replication. Whilst template unwinding is unaffected, leading strand synthesis is inhibited, leading to fork uncoupling. Synthesis through hairpin-forming repeats is rescued by replisome-intrinsic mechanisms, whereas synthesis of quadruplex-forming repeats requires an extrinsic accessory helicase. DNA-induced fork stalling is mechanistically similar to that induced by leading strand DNA lesions, highlighting structure-prone repeats as an important potential source of replication stress. Thus, we propose that our understanding of the cellular response to replication stress may also be applied to DNA-induced replication stalling.

[1] Genome Replication lab, Division of Cancer Biology, Institute of Cancer Research, Chester Beatty Laboratories, 237 Fulham Road, London SW3 6JB, UK.
[2] These authors contributed equally: Corella S. Casas-Delucchi, Manuel Daza-Martin. ✉email: Gideon.coster@icr.ac.uk

Faithful and accurate chromosomal DNA replication is a fundamental process that is required to maintain genome stability and is performed by a multi-protein complex termed the replisome[1]. The replisome encounters various types of challenges, including DNA damage, DNA-bound proteins, collisions with the transcriptional machinery, RNA-DNA hybrids (R-loops), topological stress and limiting dNTPs[2]. Under unperturbed conditions, leading strand synthesis is coupled to unwinding, and this contributes to maximal fork rates[3]. However, when synthesis is stalled, CMG can continue to unwind at a reduced rate, a scenario termed helicase-polymerase uncoupling[4–6].

In addition to exogenous factors, certain DNA sequences can intrinsically pose a challenge to the replisome, in terms of both fidelity and dynamics. Most of our understanding of how DNA affects its own replication stems from studies of expansion-prone repeats which drive nearly 50 different neurodegenerative diseases[7–9]. Roughly half of these conditions are caused by expansion of just three repeat classes – $(CGG/CCG)_n$, $(GAA/TTC)_n$ and $(CTG/CAG)_n$ (hereafter referred to as $(CGG)_n$, $(GAA)_n$ and $(CTG)_n$). In these diseases, the number of repeat units is highly variable within the general population. When repeats expand to an intermediate range, individuals may exhibit partial phenotypes. Further expansion, usually within one generation, leads to a full mutation. For example, Fragile X syndrome is caused by expansion of $(CGG)_n$ repeats within the 5′ UTR of the FMR1 locus. Unaffected individuals harbour 6–52 $(CGG)_n$ repeats, an intermediate range is 53–250, whereas patients harbour 250–2000 repeat units.

One of the earliest proposed mechanisms for contractions or expansions of repeats was replication slippage[10,11], a process by which the template and nascent strands reanneal out of register due to the repetitive nature of the template[12]. However, large scale contractions and expansions cannot easily be explained by slippage. Furthermore, slippage can occur in any repetitive sequence, yet only some repeats undergo large scale expansions. Replication, transcription and various repair pathways have been implicated in large scale expansions[13], but the exact underlying mechanisms are not yet fully understood. Current models are based on the finding that repeat expansions correlate with the propensity of sequences to fold into unusual DNA secondary structures.

Several types of non-B-DNA secondary structures have been characterised, including (i) hairpins, (ii) G-quadruplexes (G4s), (iii) intercalated motifs (i-motifs) and (iv) triplexes. Hairpins are simple intramolecular fold-back structures that rely on classic Watson–Crick base pairing. Inverted repeats and palindromes can form perfectly annealed hairpins while $(CNG)_n$ repeats can form mismatch-containing hairpins. G4s are four stranded intra- or intermolecular structures formed by Hoogsteen base pairing between guanine residues[14]. Four guanines can form a planar arrangement termed a G-quartet and stacking of multiple G-quartets yields a G4. While G-rich sequences can form G4s, C-rich sequences can form a four-stranded structure called an i-motif, where pairs of hemi-protonated cytosines form base pairing in a criss-cross pattern[15]. Hairpins, G4s and i-motifs can all form locally within a stretch of single stranded DNA (ssDNA). In contrast, triplex DNA requires a donor duplex DNA, with a third strand annealing via Hoogsteen base pairing[16]. Triplexes can arise from homopurine-homopyrimidine mirror repeats, such as $(GAA)_n$ repeats, and their formation is favoured by negative supercoiling[17].

The first evidence that repeats can stall fork progression in vivo was the detection of replication intermediates of plasmids containing $(CGG)_n$ repeats in bacteria by two-dimensional (2D) gel electrophoresis[18]. Stalling was observed in both orientations and was later also detected in budding yeast and mammalian cells[19,20]. In contrast, stalling by $(GAA)_n$ repeats in yeast only occurs when they are on the lagging strand template[21–23], whereas stalling by $(CTG)_n$ repeats is significantly weaker and is orientation independent[19,24]. Fibre labelling of individual replication forks in the CGG-expanded FMR1 locus from Fragile X syndrome cells revealed stalling[25]. Interestingly, replication forks progressed in either direction in cells from unaffected individuals, whereas almost all forks in patient cells replicated $(CGG)_n$ as the leading strand template. Similar experiments with cells from Friedreich's ataxia patients showed pronounced stalling in the GAA-expanded FXN locus. Fork directionality was also altered, positioning the $(GAA)_n$ repeats on the leading strand template[26], which is the exact opposite orientation that generates stalls in budding yeast[21–23]. The reasons for these discrepancies are unclear. Furthermore, the underlying mechanism of repeat-induced stalling is poorly defined. Stalling could be induced indirectly, for example by DNA-bound proteins or R-loops. In the case of $(CTG)_n$ repeats, stalling was suggested to be driven by binding of mismatch repair factors to mismatched hairpins[24]. This raises the question of whether the DNA template by itself is sufficient to stall the replisome. If so, which sequences stall and what is the underlying mechanism? Finally, how does the replisome recover from such blocks?

Studies of repeat replication in vitro have thus far been limited to primer extension assays and have shown that polymerases are impeded by $(CGG)_n$, $(CTG)_n$ and $(GAA)_n$ repeats[27–32]. Most studies employed bacterial or viral polymerases, with very little work done with all three eukaryotic replicative polymerases. One study compared yeast pol δ with human pols α and ε, all of which were stalled by $(CGG)_7$[33]. One limitation of such assays is the use of ssDNA templates that are pre-folded into structures. Whether sufficient ssDNA could be exposed for structures to form during unperturbed coupled leading strand synthesis is unknown. Another caveat is the lack of any additional replisome components. Reconstituted Escherichia coli replisomes are not affected by $(CTG)_n$ repeats but are stalled by short $(CCG)_n$ repeats and inverted repeats[34,35]. To date, studies of repeat replication with reconstituted eukaryotic replisomes are lacking.

In this study we set out to determine the molecular events that transpire when the eukaryotic replisome encounters repetitive templates. Using reconstituted replisomes assembled from purified budding yeast proteins, we found that certain repeats induce leading strand stalling. Since these experiments lack components from other pathways, they indicate that DNA alone can cause replication fork stalling. We tested a wide range of mono-, di- and trinucleotide repeats and found that stalling correlated most with structure-forming capacity. Mechanistically, the CMG helicase was able to continue unwinding but synthesis was inhibited, resulting in helicase-polymerase uncoupling, thereby resembling events induced by a leading strand DNA lesion. We found that the two major replicative polymerases, pols δ and ε, exhibit different inherent capacities to synthesise through hairpin-forming repeats and uncovered a role for pol δ in rescuing DNA-induced leading strand stalling. Moreover, fork recovery mechanisms differed by the type of secondary structure that repeats can form. Leading strand synthesis through hairpin-forming repeats was modulated by various replisome-intrinsic aspects, including the presence of pol δ, synthesis rate by pol ε and levels of dNTPs. In contrast, quadruplex-forming repeats were not affected by any of these factors, but instead required the extrinsic accessory helicase Pif1 for efficient replication. Altogether, these results provide a mechanistic understanding of how the eukaryotic replisome copes with challenging repetitive templates and highlights certain sequences as an important potential source of endogenous replication stress.

## Results

**$(CGG)_n$ repeats induce leading strand stalling**. To investigate the effect of repeats on the eukaryotic replisome, we constructed a set of substrates for in vitro replication assays, whereby eukaryotic replisomes are assembled using purified budding yeast proteins[3,36]. $(CTG)_n$, $(GAA)_n$ and $(CGG)_n$ repeats were cloned 3 kb downstream of the replication origin (Fig. 1A) of a 9.7 kb substrate that supports origin-specific replication initiation[5]. Short oligonucleotides were used for initial cloning, followed by a PCR-free approach which involved iterative steps of controlled expansion of repeats to yield substrates with up to 161 uninterrupted repeats[37]. Given the potentially unstable nature of certain repeats during propagation in bacteria, we validated that our final preparations contained the correct insert size and sequence (Supplementary Fig. 1). Since replication initiates from a defined position, we can assign which sequences serve as the leading and lagging strand templates. When describing insert sequences throughout this manuscript, we refer to sequences that reside on the leading strand template. For example, the $(CGG)_{61}$ substrate contains 61 CGG repeats on the leading strand template, and therefore 61 CCG repeats on the lagging strand template.

To avoid the confounding effects of two replication forks converging on a circular template, we first performed reactions on linear templates. Plasmids were linearised with a restriction enzyme (AhdI) such that the replication origin was positioned 1.5 kb from one end, and 8.2 kb from the other, with the repeats located within the 8.2 kb fragment. Enzymes required for Okazaki fragment maturation were omitted to simplify analysis. As expected, analysis of the control replication reaction by denaturing alkaline gel electrophoresis produced three main products: the leftward moving 1.5 kb leading strand, the rightward moving 8.2 kb leading strand, and a heterogeneous population of smaller unligated lagging strand Okazaki fragments (Fig. 1B, lane 1). Replication of substrates containing $(CTG)_{161}$ did not differ from the empty vector control (Fig. 1B, lanes 1–2). However, a very faint 3 kb stall band was reproducibly detected with $(GAA)_{161}$ and $(CGG)_{161}$ (Fig. 1B, lanes 3,4). The intensity of this stall was increased when reactions were performed without pol δ for $(CGG)_{161}$ but not for $(GAA)_{161}$ (Fig. 1C). This suggests a role for pol δ in preventing or rescuing leading strand stalls induced by $(CGG)_{161}$. Since these experiments lack components from other pathways, we conclude that the DNA template itself can induce fork stalling, and that this is modulated by polymerase usage.

**Stalling threshold is 17 $(CGG)_n$ repeats and is orientation-dependent**. Although both $(GAA)_{161}$ and $(CGG)_{161}$ produced leading strand stalls, further analysis of $(GAA)_{161}$ stalls proved difficult due to the weak signal. We therefore focused on $(CGG)_n$-induced stalls, which were sufficiently robust when pol δ was absent. To establish the threshold for $(CGG)_n$ stalling we replicated a set of substrates with increasing repeat units in the absence of pol δ. This revealed that as few as 17 repeats were sufficient to induce some stalling, which was further enhanced with 21 and 41 repeats, and saturated with 41 repeats or more (Fig. 1D, see also quantification of five independent experiments in Supplementary Fig. 2a). Similar results were obtained with circular plasmids in the presence of topoisomerase I (Supplementary Fig. 2b–d), indicating that stalling is neither promoted nor prevented by a topologically closed template or by topoisomerase activity. When compared to a stall driven by a site-specific leading strand DNA lesion (a cyclobutane pyrimidine dimer; CPD), even the longest $(CGG)_n$ inserts produced a partial stall, also evident by the larger proportion of full length 8.2 kb products (Fig. 1D, compare lanes 9 and 10). Consistent with the

accumulation of stalled forks, large replication intermediates were observed by native gel electrophoresis (Supplementary Fig. 2e), mirroring the pattern seen by alkaline denaturing analysis. We note that $(CGG)_n$ inserts containing 81 repeat units or more were not completely stable in bacteria (Supplementary Fig. 1b, lanes 7–10, seen as smearing below the main band). We therefore chose to use $(CGG)_{61}$ in all subsequent experiments as it drove maximal stalling but was genetically stable.

If, as suggested by genetic evidence, the orientation of repeats relative to replication origins plays a role, one might expect to observe a difference in stalling as a function of orientation. To test this idea, we reversed the orientation of these repeats to yield $(CAG)_n$, $(TTC)_n$ and $(CCG)_n$ templates. While we were able to clone $(CAG)_{161}$ and $(TTC)_{161}$, we were only able to obtain stable clones of up to 61 CCG repeat units, as longer CCG repeats are unstable in this orientation in bacteria[38,39]. Nonetheless, in contrast to $(CGG)_n$ templates, replication of all $(CCG)_n$ substrates produced no detectable stalls (Fig. 1E), even when compared side-by-side (Supplementary Fig. 2f). Replication of $(CAG)_{161}$ and $(TTC)_{161}$ produced no stalling with either linear or circular templates (Supplementary Fig. 2g, h). In summary, as many as 161 $(CTG)_n$ or $(GAA)_n$ repeats do not induce robust replication stalling in either orientation, whereas 17 $(CGG)_n$ repeats or more do so, but only when positioned on the leading strand template.

**Short $(CG)_n$ repeats also induce leading strand stalling**. The fact that $(CGG)_n$ produced pronounced stalling, yet other trinucleotide repeats did not, suggested that it is not simply their repetitive nature that causes a stall. We considered the possibility that stalling is caused by DNA secondary structures. While all $(CNG)_n$ repeats can fold into hairpins, the thermal stability of $(CGG)_n$ hairpins is significantly higher[40], possibly explaining the stalling observed only with $(CGG)_n$. This raises the prediction that other G-rich hairpin-forming repeats may also stall the replisome. To test this, we cloned and replicated a range of dinucleotide repeats. Of these, stalling was only observed with $(CG)_n$ repeats (Fig. 2A and Supplementary Fig. 3a), which are indeed G-rich and form hairpins in solution[32]. Relative to $(CGG)_n$, much shorter stretches of $(CG)_n$ dinucleotides produced a strong stall (Fig. 2A), with a lower threshold of only 10 repeat units. Similar to that observed with $(CGG)_n$ templates, analysis of $(CG)_n$ replication products on a native gel revealed accumulation of replication intermediates (Supplementary Fig. 3b) and the stalling threshold was similar with circular templates (Supplementary Fig. 3c). Another important class of hairpin-forming dinucleotides are $(AT)_n$ repeats. Long $(AT)_n$ repeats ($n = 34$) interfere with replication and cause chromosome fragility in budding yeast[41] and are expanded in microsatellite unstable cancers[42]. Despite much effort, we were not able to generate $(AT)_n$ repeats longer than 15 units, leaving open the question of whether long $(AT)_n$ repeats can stall the replisome in vitro.

Our results provide further evidence that hairpin-forming repeats can stall the replisome. To further support this interpretation, we generated scrambled sequences with the same length, base pair composition and strand bias as $(CGG)_{21}$ or $(CG)_{24}$. For each repeat type we chose two randomly generated sequences which contain minimal stretches of consecutive CG or CGG repeats, thereby interrupting continuous base-pairing within the predicted hairpins. All the scrambled sequences were replicated without any stalling (Fig. 2B, C). Altogether, these results indicate that the nucleotide composition and strand bias of $(CGG)_n$ and $(CG)_n$ repeats do not account for their ability to stall leading strand synthesis. Rather, stalling is most consistent with their structure-forming potential.

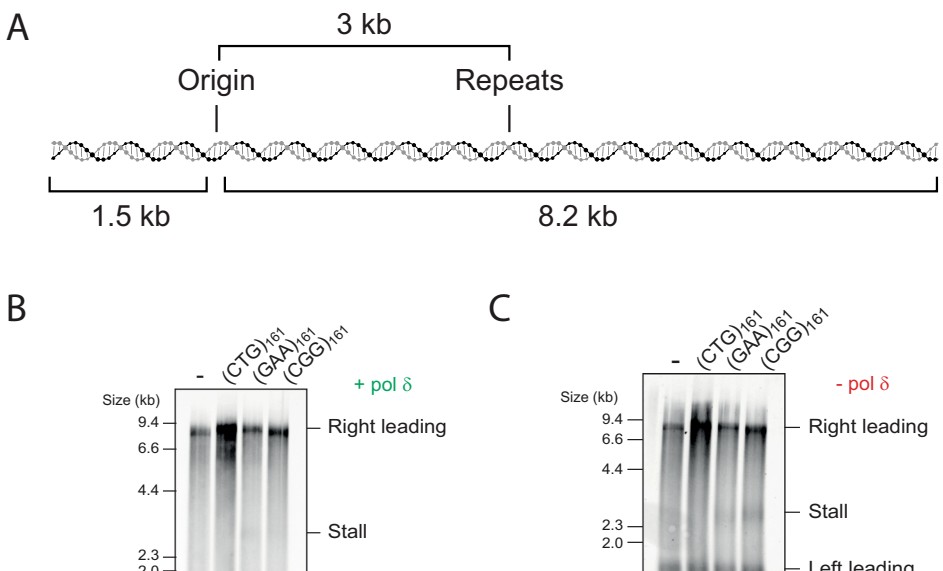

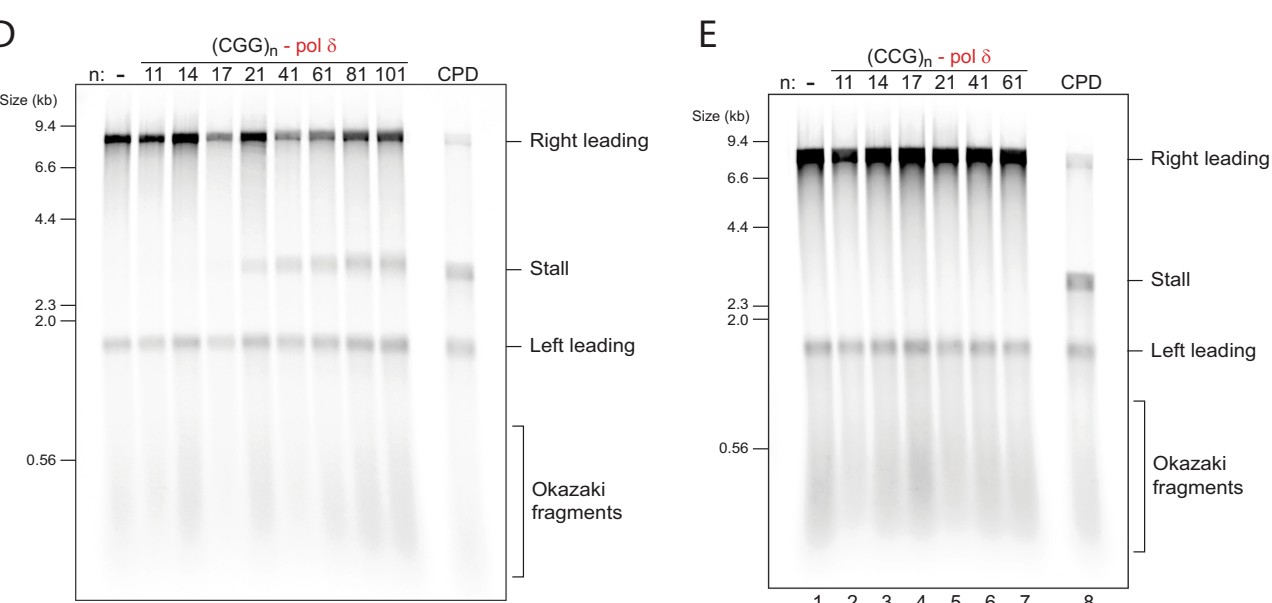

**Fig. 1 (CGG)$_n$ repeats induce orientation-dependent leading strand stalling. A** Schematic of the replication substrate used in this study. The ARS306 origin of replication drives sequence-specific initiation events. Two forks emanating from the origin generate a leftward moving 1.5 kb leading strand ("Left leading") and a rightward moving 8.2 kb leading strand ("Right leading"). Various repeats were cloned 3 kb downstream such that only the rightward moving fork would encounter them. Thus, the left leading product serves as an internal control. Analysis of in vitro replication reactions by denaturing gel electrophoresis, using linear substrates with different types of leading strand repeats, in the presence (**B**) or absence (**C**) of pol δ. Replication reactions carried out in the absence of pol δ with a series of substrates containing increasing numbers of either (CGG)$_n$ (**D**) or (CCG)$_n$ (**E**) leading strand repeats, as well as a comparison to a site-specific leading strand CPD.

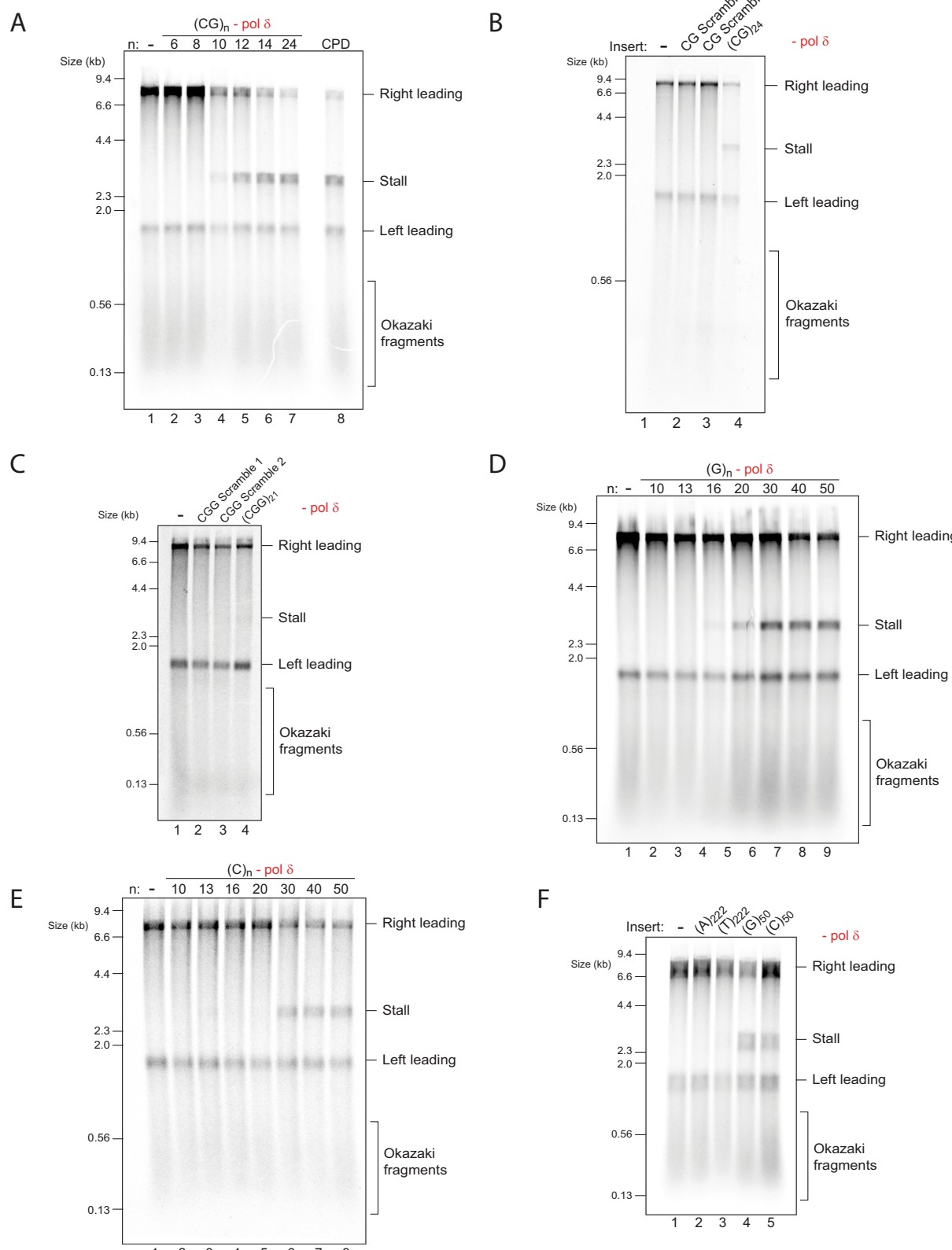

**Fig. 2 (CG)ₙ, (G)ₙ and (C)ₙ also induce leading strand stalling. A** Replication reactions carried out in the absence of pol δ with a series of substrates containing increasing numbers of (CG)$_n$, as well as a comparison to a site-specific leading strand CPD. Replication reactions carried out in the absence of pol δ, comparing two randomly generated scrambled sequences with the same base-pair composition and strand bias as either (CG)$_{24}$ (**B**) or (CGG)$_{21}$ (**C**). Replication reactions carried out in the absence of pol δ with a series of substrates containing increasing numbers of either guanine (**D**) or cytosine (**E**) leading strand homopolymers. **F** Replication reactions carried out in the absence of pol δ with a series of substrates containing different leading strand homopolymer templates as indicated.

**The replisome is affected by quadruplex-forming homopolymers**. The leading strand stalling we observed correlated with the ability of sequences to fold into hairpin structures. We reasoned that repeats that form other types of DNA secondary structures may also impede replication. We therefore tested the effect of guanine and cytosine homopolymers, which can fold into a G4 or i-motif, respectively[32]. Leading strand stalls were indeed observed, with a threshold of 20 and 30 repeat units for $(G)_n$ and $(C)_n$, respectively (Fig. 2D, E). This difference in threshold was also seen when compared side-by-side within the same experiment (Supplementary Fig. 3d) and was maintained on circular templates (Supplementary Fig. 3e, f). In contrast, stretches of over 200 consecutive adenine or thymine residues, which are not predicted to form stable secondary structures, did not cause a significant stall (Fig. 2F). Altogether, we conclude that hairpin- and quadruplex-forming repeats can stall the replisome.

**Pol δ drives recovery from hairpin-forming, but not quadruplex-forming repeats**. Our results thus far highlight four different types of repeats that induce leading strand stalling— $(CGG)_n$, $(CG)_n$, $(C)_n$ and $(G)_n$. Given our initial observation that pol δ can assist replication through $(CGG)_{161}$ (Fig. 1B, C), we next asked whether this holds true for the other sequences. While replication of $(CGG)_{61}$ and $(CG)_{24}$ was improved by the presence of pol δ, stalling by $(C)_{50}$ and $(G)_{50}$ was essentially unaffected (Fig. 3A and Supplementary Fig. 3g). Thus, the ability of pol δ to synthesise past these sequences correlates with the type of secondary structure that they can form.

To assess if stalling is terminal or transient, we performed pulse-chase experiments, in which nascent DNA was labelled with dATP for the first 10 minutes and chased with excess unlabelled dATP, allowing us to follow the fate of forks labelled within the pulse without detection of new initiation events. In the absence of pol δ, stalling by $(CGG)_{61}$ was persistent for at least two hours, indicating that pols α and ε are unable to resolve this stall (Fig. 3B). In contrast, in the presence of pol δ stalling at the earliest time point was weaker, gradually resolved over time, and was barely discernible by 40 minutes (Fig. 3C). A similar pattern was observed with $(CG)_{24}$ (Fig. 3D, E). These results indicate that pol δ does not prevent the formation of stalls induced by $(CGG)_n$ and $(CG)_n$ but rather resolves them. Pulse chase experiments with $(C)_{50}$ and $(G)_{50}$ revealed persistent stalling regardless of the presence of pol δ (Supplementary Fig. 4), further supporting our earlier observation that pol δ cannot support replication through these two sequences (Fig. 3A). In summary, hairpin-forming sequences induce persistent stalls in the absence of pol δ, but these are resolved over time when pol δ is present. In contrast, G4- and i-motif-forming sequences generate persistent stalls that cannot be resolved by pol δ.

The ability of pol δ to rescue certain leading strand stalls could either require its continued presence within the replisome during stalling or could occur behind the fork. To test whether pol δ could rescue pre-existing stalls we carried out pulse-chase experiments in which stalls were pre-formed during the pulse, and pol δ was only introduced in the chase. Figure 3F shows that a clear stall with $(CG)_{24}$ was evident after the 10 min pulse (lane 1), which remained unaltered in the absence of pol δ for 30 min (lanes 2-6). Rescue by pol δ was observed within 2.5 min (Fig. 3F, lane 7), showing similar rescue kinetics with stalls induced by $(CGG)_{61}$ (Supplementary Fig. 5a). Leading strand rescue by pol δ was largely dependent on RFC/PCNA (Supplementary Fig. 5b), suggesting that PCNA is either retained or reloaded on the leading strand template. Importantly, adding PCNA in the absence of pol δ had no effect. We conclude that pol δ can rescue pre-existing leading strand stalls in a PCNA-dependent fashion.

**DNA-induced stalls trigger helicase-polymerase uncoupling**. Replication forks could either stall due to impaired unwinding by the CMG helicase or inhibition of synthesis by pol ε, which would trigger uncoupled unwinding downstream of the stall. The fact that pol δ could rescue pre-existing stalls (Fig. 3F) supports the latter, as it strongly argues for the presence of a free primer-template junction and an available exposed template downstream. Previous work revealed that repriming past a leading strand CPD by pol α is inefficient, and that an exogenously added primer allows resumption of leading strand synthesis[5]. Primer annealing only occurs if ssDNA is exposed, thereby serving as an indirect measure of uncoupled CMG unwinding. We therefore asked whether a primer that anneals 265 nt downstream of the insert would promote the formation of a restart product. Indeed, addition of this primer, but not a scrambled control primer, led to the appearance of a 5 kb restart product for all four stall-forming repeats, to an extent similar to that seen with a leading strand CPD template (Fig. 4A). This result strongly suggests that stalling is not a consequence of CMG arrest, but is rather due to lack of synthesis by pol ε. Interestingly, while pol δ resolved the 3 kb stall products induced by $(CGG)_{61}$ and $(CG)_{24}$, 5 kb restart products were still evident (Fig. 4B, lanes 8 and 9). Therefore, CMG continued to unwind at least 265 nt beyond the repeats in both cases. Thus, although pol δ can resolve certain leading strand stalls, it cannot completely prevent uncoupling.

Additional evidence for helicase-polymerase uncoupling was seen upon closer inspection of replication products analysed on native gels, whereby faster migrating species accumulated. These species were previously shown to correspond to uncoupled products, in which CMG has unwound to the end of the template but without any synthesis[5]. This was especially clear with the $(CG)_n$ templates, where uncoupled products accumulated at levels similar to those observed with a CPD containing template (Supplementary Fig. 3b). To increase the fraction of uncoupled products, we truncated substrates with EcoRV so that CMG has to unwind only 1.6 kb beyond the insert rather than 5 kb. When analysed on a native gel, uncoupled products were observed for all four classes of sequences (Fig. 4C) but were not observed for $(CGG)_{61}$ and $(CG)_{24}$ when pol δ was present (Fig. 4D). Based on the repriming experiment (Fig. 4B), it is likely that there was some degree of transient uncoupling. Pol δ was then able to synthesise past the repeats, which converted the uncoupled product into a full-length product. Altogether, these results show that structure-forming repeats can trigger helicase-polymerase uncoupling and that pol δ limits the extent of uncoupling by rescuing leading strand synthesis at $(CGG)_{61}$ and $(CG)_{24}$, but not at $(C)_{50}$ or $(G)_{50}$.

**Read-through of $(CGG)_n$ and $(CG)_n$ is facilitated by pol ε variants or elevated dNTPs**. The observation that pol ε could not synthesise past $(CGG)_n$ or $(CG)_n$, yet pol δ could, may be explained by their different enzymatic properties. More specifically, the weak strand displacement activity of pol ε relative to pol δ might preclude it from coping with hairpin-forming repeats. This activity can be mildly enhanced by inactivating the exonuclease domain of pol ε[43]. In addition, modelling of the most frequent cancer-associated pol ε mutation (P286R) in budding yeast (P301R) revealed a hyperactive enzyme in which DNA entry into the exonuclease domain is blocked, allowing it to synthesise past a hairpin structure more efficiently than an exonuclease-dead mutant[44,45]. We therefore wondered whether these pol ε variants might be able to resolve leading strand stalls even in the absence of pol δ. Leading strand stalls induced by $(CGG)_{61}$ were significantly weaker in reactions carried out with either pol ε mutants relative to WT pol ε (Fig. 5A, lanes 1–6). Stalling

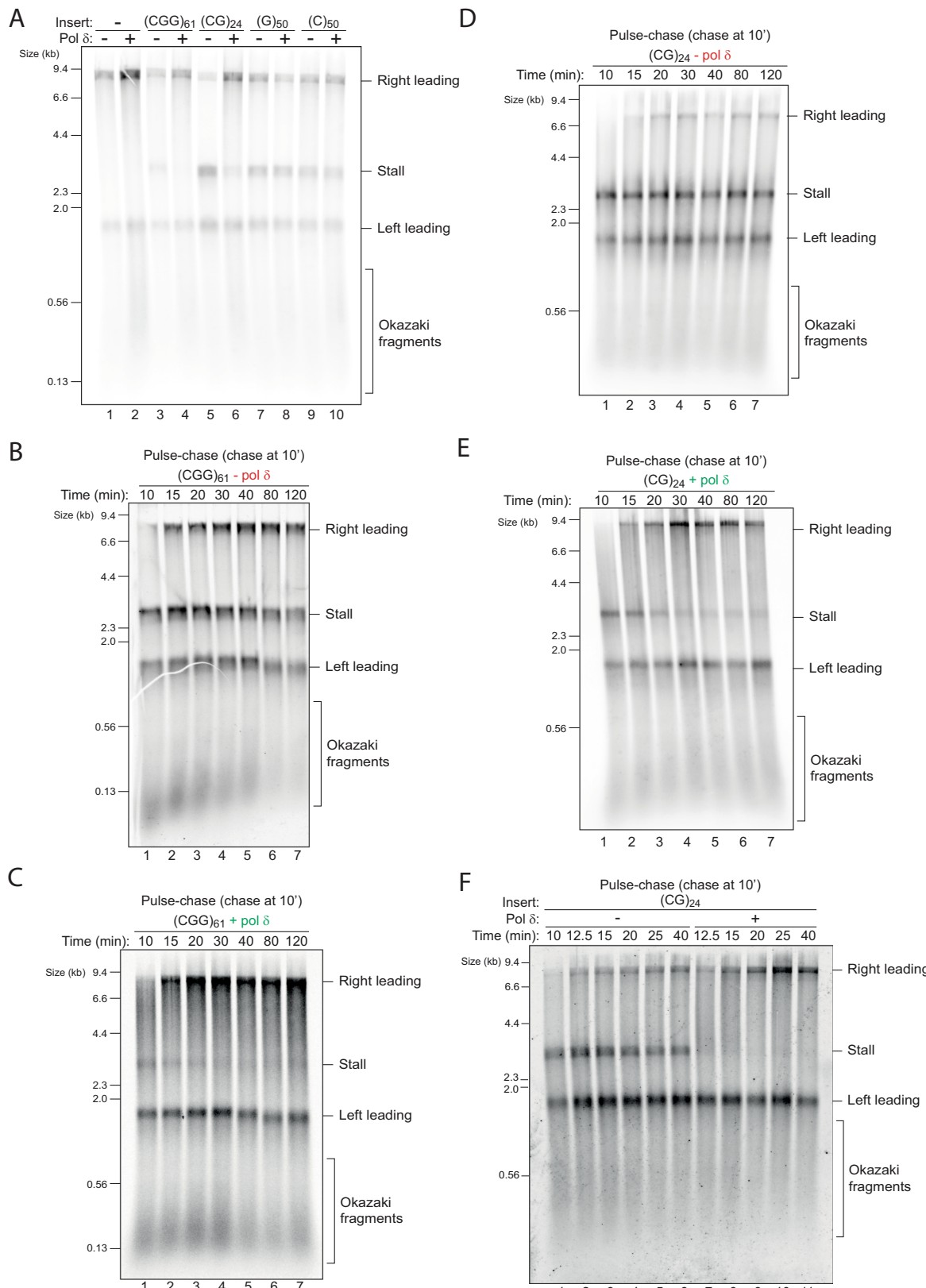

**Fig. 3 Pol δ drives recovery from (CG)$_{24}$ and (CGG)$_{61}$ stalls, but not (G)$_{50}$ or (C)$_{50}$. A** Replication reactions carried out with the indicated templates in the absence or presence of pol δ. Pulse-chase experiments carried out with the (CGG)$_{61}$ template in the absence (**B**) or presence (**C**) of pol δ. Reactions were initiated with radiolabelled dATP for 10 min, chased with excess 'cold' dATP and samples taken at the indicated time points. **D**, **E** Same as in **B**, **C** but with the (CG)$_{24}$ template. **F** Pulse-chase experiments carried out with the indicated template. Reactions were initiated in the absence of pol δ to generate a pre-existing stall. After a 10 min pulse, pol δ was either added with the chase or not and samples taken at the indicated time points.

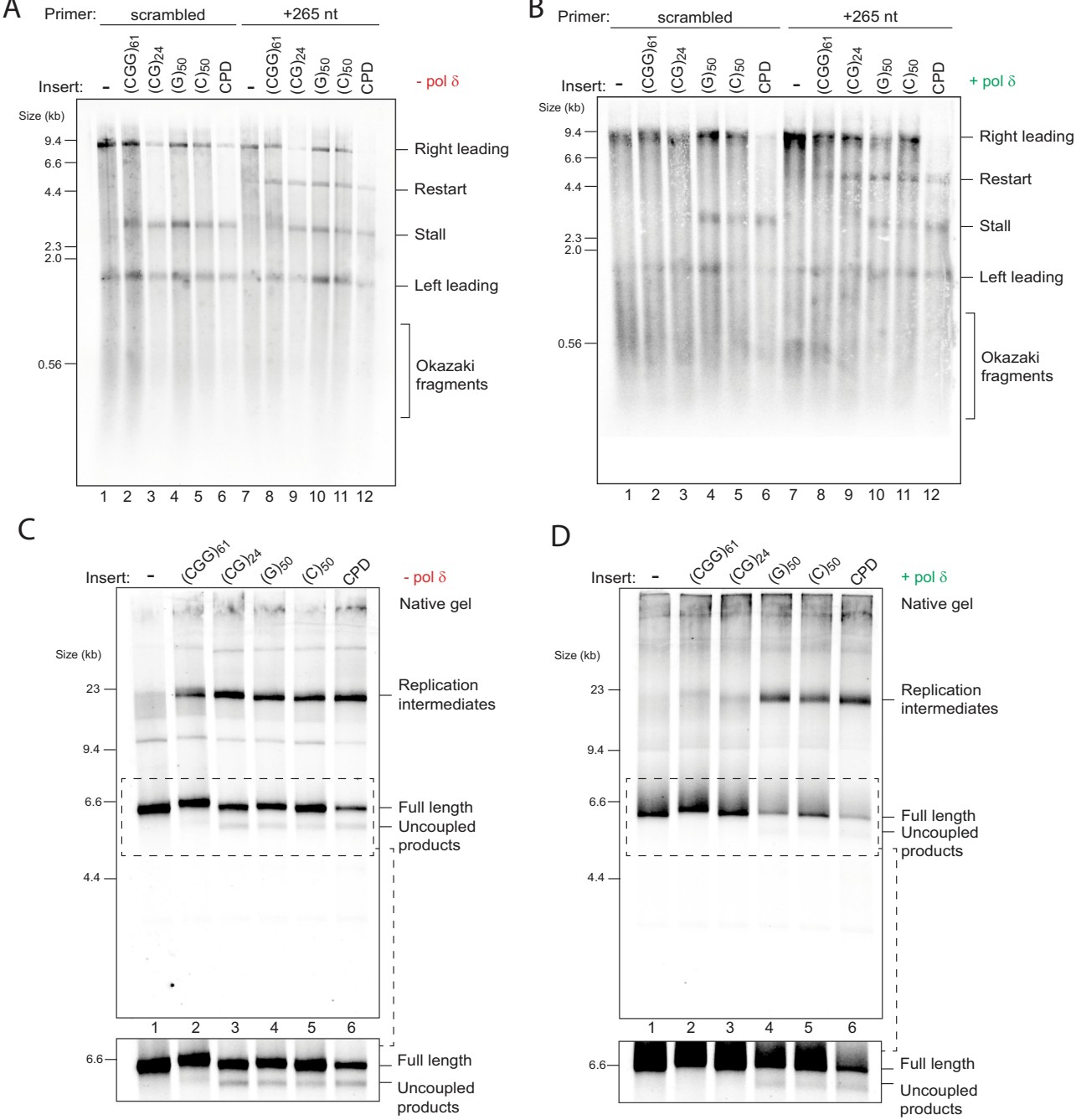

**Fig. 4 DNA-induced stalls trigger helicase-polymerase uncoupling.** Replication reactions carried out with the indicated templates in the presence of a primer that anneals 265 nt downstream of the repeats or a scrambled control primer in the absence (**A**) or presence (**B**) of pol δ. Replication reactions carried out with the indicated templates in the absence (**C**) or presence (**D**) of pol δ analysed by native gels. The insets show a longer exposure of the regions bound by the dashed boxes to better visualise uncoupled products.

produced by $(CG)_{24}$ was also rescued, but here P301R performed better than Exo- (Fig. 5A, lanes 7–12, also see quantification of five independent experiments). In contrast, neither of these pol ε variants were able to replicate past $(G)_{50}$ or $(C)_{50}$ (Supplementary Fig. 6a) and similar results were obtained when pol δ was present (Supplementary Fig. 6b). Pol ε P301R was able to rescue pre-existing stalls produced by WT pol ε (Supplementary Fig. 6c), and this was largely dependent on RFC/PCNA (Supplementary Fig. 6d). These observations are almost identical to those obtained with pol δ (Fig. 3F and Supplementary Fig. 5b), suggesting that pol ε P301R and pol δ employ a similar mechanism to rescue leading strand stalls.

Inactivation of the exonuclease domain of pol ε shifts the balance from proofreading to synthesis, leading to an overall increase in synthesis rate. Other factors that enhance synthesis rate could also play a role. We therefore asked whether increased dNTPs could ameliorate DNA-induced stalling. We performed pulse-chase experiments in which dATP was the labelled nucleotide, and chased with either excess unlabelled dATP alone, or an excess of all four dNTPs (raised from 30 μM to 400 μM). In the absence of pol δ, elevated dNTPs significantly improved replication past $(CGG)_{61}$ but not $(CG)_{24}$ (Fig. 5B, compare lanes 5 vs 6 and 8 vs 9). In the presence of pol δ, excess dNTPs also improved synthesis past $(CG)_{24}$ (Fig. 5B, compare lanes 17 vs 18).

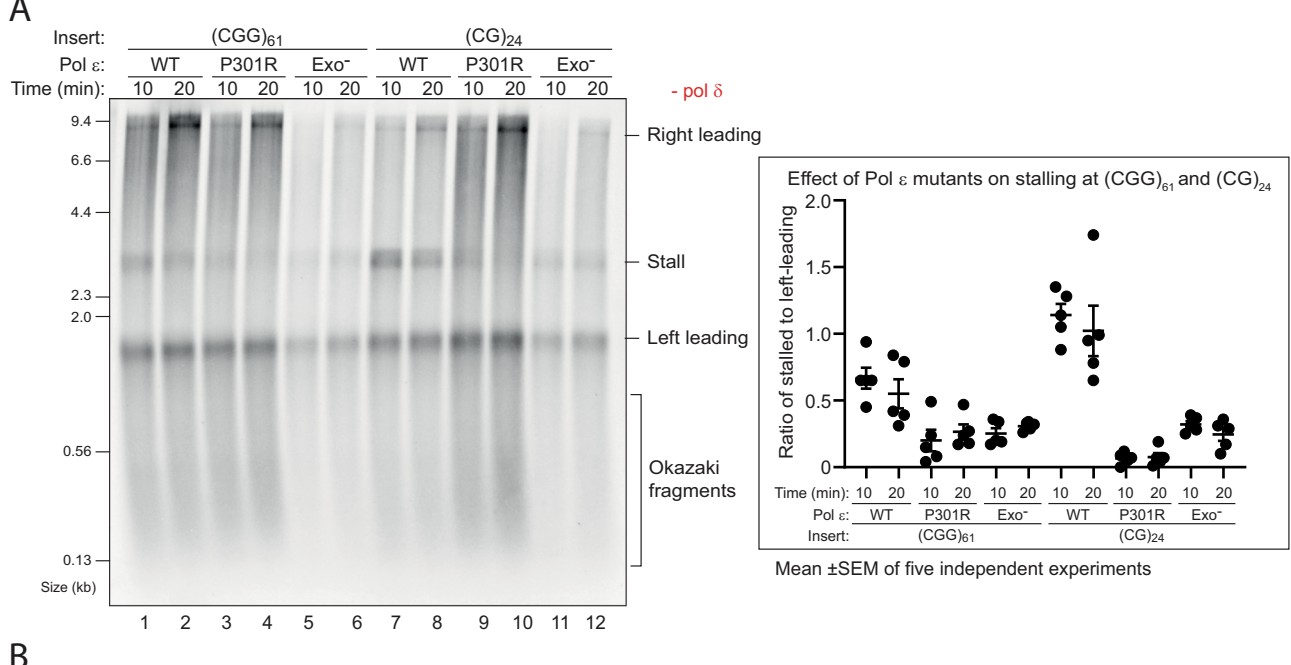

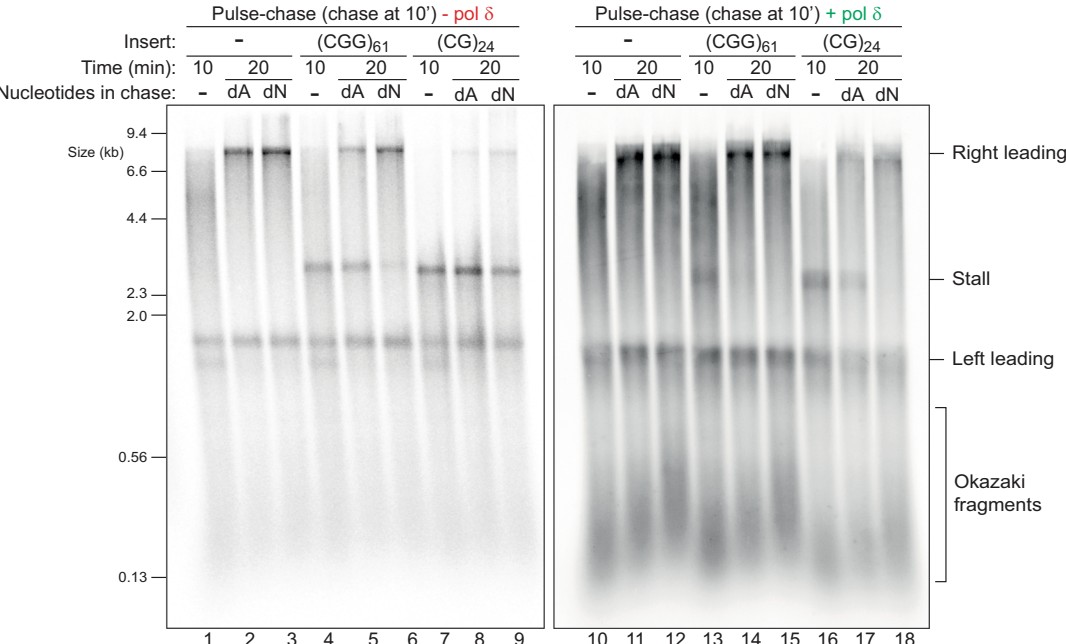

**Fig. 5 Read-through of (CGG)$_n$ and (CG)$_n$ is facilitated by pol ε variants or elevated dNTPs. A** Replication reactions carried out with the indicated templates with different pol ε variants in the absence of pol δ. Quantification shown on the right is from five independent experiments. Shown is the mean, error bars are standard deviation. **B** Pulse-chase experiments carried out with the indicated templates in the absence (left panel) or presence (right panel) of pol δ. Reactions were initiated with radiolabelled dATP for 10 min and chased for another 10 min with either excess 'cold' dATP alone (dA) or with excess of all four dNTPs (dN).

In contrast, increased dNTPs had no effect on replication of (G)$_{50}$ or (C)$_{50}$, regardless of pol δ (Supplementary Fig. 7a). Thus, increased concentrations of dNTPs improve the ability of both replicative polymerases to resolve stalls induced by hairpin-forming repeats. Combined with the results obtained with pol δ and pol ε variants, we conclude that the replisome can cope with hairpin-forming repeats by a variety of replisome-intrinsic mechanisms.

**Pif1 resolves DNA-induced stalls**. In contrast to hairpin-forming repeats, none of the conditions or enzyme variants we tried thus far allowed the replisome to cope with stalls induced by (G)$_{50}$ and (C)$_{50}$, both of which can form quadruplex structures. We considered that the ssDNA binding protein RPA may play a protective role as it has been demonstrated to unfold G4 structures[46–48]. However, stalled products were observed across a broad range of RPA concentrations (10–200 nM),

regardless of pol δ (Supplementary Fig. 7b, c and Supplementary Fig. 8). Therefore, in this context, RPA does not prevent or resolve DNA-induced leading strand stalls.

Several accessory helicases have been implicated in replication of repetitive or structure-prone DNA[49,50]. In budding yeast, Pif1 has been shown to play an important role in allowing efficient replication past G4 sequences in vivo[51–53] and in vitro[54–58]. We therefore assayed the ability of purified Pif1 to rescue DNA-induced stalled forks. Strikingly, not only was Pif1 able to fully rescue replication past $(G)_{50}$, it also accelerated replication through all of the other sequences (Fig. 6A). Importantly, an ATPase active site mutant of Pif1 (K264A) which cannot unwind DNA (Supplementary Fig. 9a), was unable to perform any of these tasks (Fig. 6), indicating an essential requirement for its helicase motor function. For comparison, we also tested the nuclease-helicase Dna2, but found it had no effect on DNA-induced stalling despite showing robust nuclease activity (Supplementary Fig. 9b, c). Pif1 was previously shown to directly bind PCNA[59] and to collaborate with pol δ and PCNA in break induced replication (BIR)[60,61] and in stimulating strand displacement during lagging strand maturation[62–64]. However, our results show that the ability of Pif1 to resolve DNA-induced stalls is distinct from these functions, as it did not require pol δ (Fig. 6B) or PCNA (Supplementary Fig. 9d). Altogether, we conclude that Pif1 is a general-purpose accessory helicase that accelerates recovery from a variety of leading strand DNA-induced stalls.

## Discussion

We have reconstituted repeat replication with eukaryotic replisomes and have found that DNA alone is sufficient to cause significant leading strand stalling. Therefore, certain DNA sequences are an important source of endogenous replication stress. Mechanistically, stalling by DNA repeats and leading strand DNA lesions share similarities—CMG unwinding is unaffected and inhibition of synthesis triggers helicase-polymerase uncoupling. Furthermore, we demonstrate that the two major replicative polymerases exhibit different inherent capacities to cope with repetitive templates, with pol δ showing more robust activity than pol ε, allowing it to rescue leading strand stalls caused by hairpin-forming repeats. The replisome could recover from stalls induced by hairpin-forming sequences by employing a variety of replisome-intrinsic mechanisms, including pol δ, hyperactive pol ε or elevated dNTPs. In contrast, stalls induced by quadruplex-forming sequences required extrinsic support, revealing a general role for the Pif1 helicase in accelerating recovery from a variety of DNA-induced stalls. These results invoke several interesting and important questions, including the root cause of stalling and the emergence of different recovery mechanisms.

It is evident that only certain sequences induce leading strand stalling, yet the underlying reason is unclear. Our results show stalling cannot be easily explained by the repetitive nature of sequences, their base pair composition or their strand bias. Rather, fork stalling is best correlated with the ability of sequences to fold into stable DNA secondary structures. Although $(CGG)_n$ repeats have been shown to fold into a G4 structure[65] or Z-DNA[66] in vitro, this only occurs under non-physiological conditions[67,68]. Thus, hairpins are the most physiologically likely structures formed by $(CGG)_n$, with stretches of over 12 repeats suggested to form branched hairpins[68]. The stall threshold we observed ($n = 17$) was surprisingly low, meaning that in most normal FMR1 alleles ($n = 5–63$) local uncoupling may occur, providing a plausible mechanism for small scale expansions.

Out of all the sequences we tested, stalling by relatively short $(CG)_n$ repeats exhibited the highest proportion of stalled forks.

Recent NMR analysis shows that in solution $(CG)_n$ repeats form hairpins[32]. Although $(CG)_n$ repeats could in theory also form cruciforms ahead of the fork, this does not happen even in negatively supercoiled plasmids[69] because CG-rich DNA inhibits cruciform nucleation[70]. Interestingly, $(CG)_n$ repeats are extremely rare - not only in the human genome, but across the entire tree of life - constituting less than 1% of all dinucleotides in most species[71]. Methylation of cytosine within CpG increases its rate of deamination, resulting in C to T transitions. This has been proposed as the main evolutionary mechanism for genomic suppression of $(CG)_n$ dinucleotides[72]. However, trinucleotides such as $(CGG)_n$ do not show such remarkable genomic depletion, despite harbouring the same CpG sequences. This suggests that $(CG)_n$ sequences undergo negative selection. We propose that the capacity of $(CG)_n$ to efficiently stall replication serves as a selective force that leads to their genomic suppression.

The weak stalling by $(GAA)_n$ repeats may seem unexpected, as these repeats induce robust stalling in vivo in multiple organisms[21–23,26]. However, the fact that stalling is observed in opposite orientations in yeast and human patient derived cells strongly points to additional factors being involved. One possible factor could be sequence context. Analysis of SV40-based $(GAA)_n$ plasmids by electron microscopy revealed the formation of unusual fork structures such as reversed forks[73]. Interestingly, only weak and transient stalling was observed. Triplex structures were also observed, and these formed between the $(GAA)_n$ repeats and other GA-rich regions within the plasmid. It is therefore possible that our substrates lack a sufficiently long second GA-rich array to serve as a dsDNA donor. An alternative explanation was raised by a recent study carried out in DT40 cells, where replication stalling by relatively short $(GAA)_n$ tracts was suggested to occur due to R-loops[74]. Finally, we cannot exclude the possibility that our reaction conditions are not conducive for triplex formation. Altogether, we conclude that within our experimental conditions, $(GAA)_n$ repeats by themselves cause mild leading strand stalling.

Our results with guanine homopolymers are consistent with previous analysis of the effects of G4 forming sequences on replication and the role of Pif1 in resolving stalling[51] and are consistent with a recent study on the interplay between R-loops and G4 formation[75]. While past work supports the idea that G4 structures impede replication, the evidence is conflicting with regards to the effect of their orientation relative to replication origins. Loss of epigenetic information in avian DT40 cells due to uncoupling can be induced by a single G4 forming sequence, but only when positioned on the leading strand template[76]. Similarly, genetic instability of G4-forming human minisatellites in budding yeast is only induced when the G-rich strand is positioned on the leading strand template[52]. In contrast, live cell imaging of fluorescent arrays in budding yeast detected delays in replisome progression only when G4 sequences were positioned on the lagging strand template[53]. Our results show that cytosine homopolymers also induce leading strand stalling and NMR spectroscopy analysis directly demonstrated that $(C)_{22}$ forms an i-motif[32]. It is therefore possible that for some G4-forming sequences the C-rich strand produces a stall due to an i-motif structure, whereas in other cases the G-rich strand does so due to a G4 structure.

It is worth noting that we have only tested a single G4-forming sequence and a single i-motif forming sequence. These homopolymers may not accurately represent how other quadruplex-forming sequences behave. Therefore, an important area of future study is to establish how other quadruplex-forming sequences affect replication.

In the context of replication, DNA secondary structures could either form ahead of the fork or behind the fork. Our current working model is that DNA secondary structures form behind

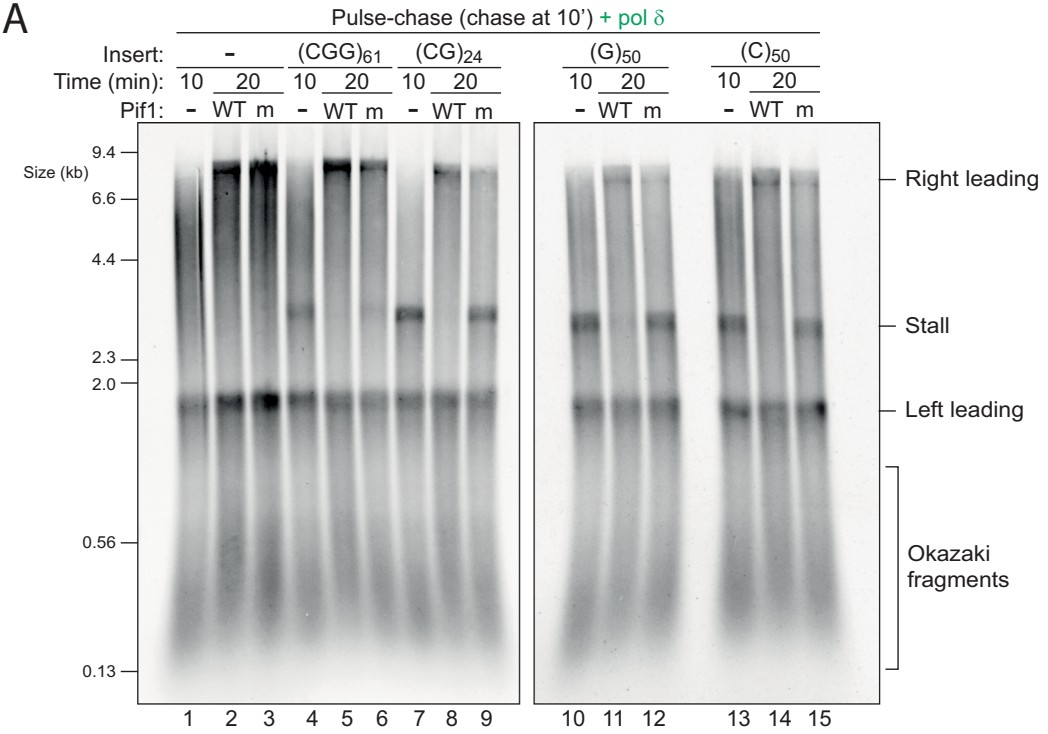

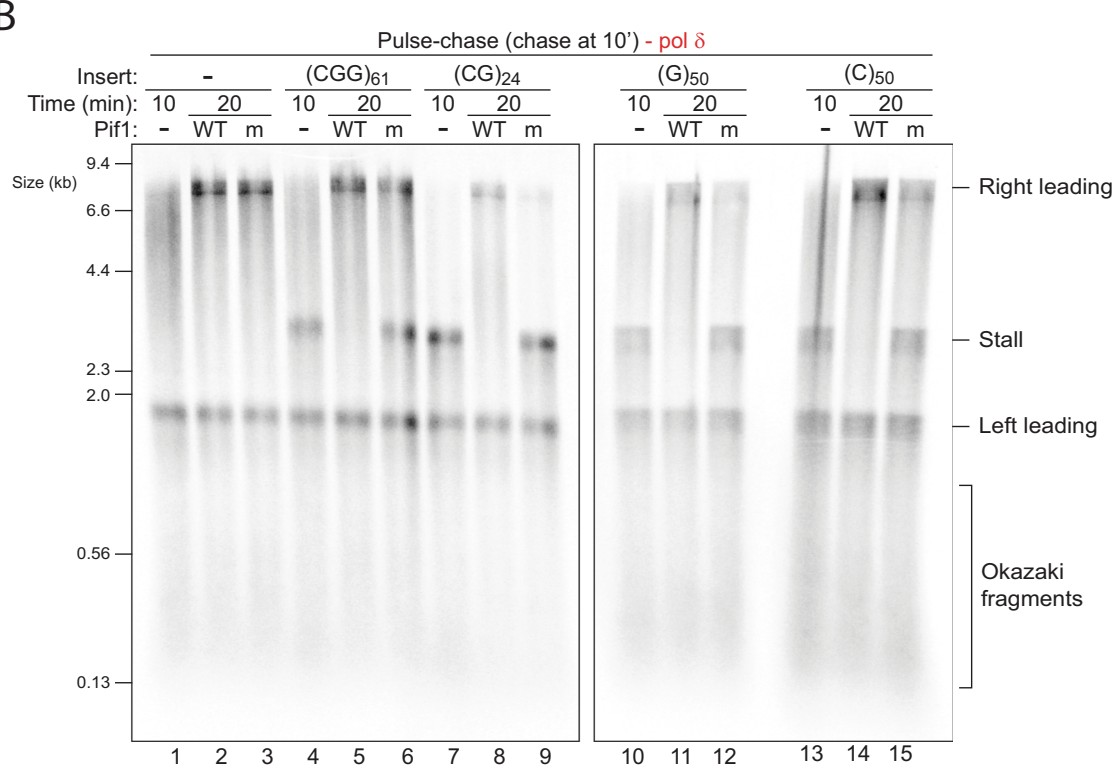

**Fig. 6 Pif1 resolves DNA-induced stalls.** Pulse-chase experiments carried out with the indicated templates in the presence (**A**) or absence (**B**) of pol δ. Reactions were initiated with radiolabelled dATP. After a 10 min pulse, either WT or ATPase-dead (K264A) pif1 was added with the chase and samples taken after another 10 min.

CMG, since we observe efficient uncoupling. However, we cannot rule out the possibility that our substrates contain pre-existing structures, and that these are bypassed intact by CMG. Recent work in Xenopus egg extracts revealed that CMG is able to bypass a large protein cross-linked to the leading strand template[77], although this required generation of ssDNA downstream by the accessory helicase Rtel1, and may require additional factors. Single molecule studies revealed that yeast CMG possesses an

Mcm10-dependent "gate" that allows it to transition from ssDNA to dsDNA[78], which could perhaps allow it to bypass certain structures on the leading strand. Alternatively, CMG may unwind and dismantle pre-existing structures. While the capacity of CMG to unwind various DNA secondary structures is unknown, our results imply that if CMG does unwind structures, these must form again behind it to inhibit synthesis. But how could structures arise on the leading strand if unwinding and synthesis are coupled? Although pol ε directly binds CMG, its catalytic domain is tethered via a flexible linker[79]. This raises the possibility that stochastic disengagement of pol ε from the leading strand template leads to local uncoupling and exposure of short stretches of ssDNA, thereby allowing structures to form. However, fork stalling induced by $(CG)_{24}$ was extensive, which would require such a stochastic event to be very frequent. Another option is that structures form on the ssDNA stretch that runs between the exit channel of CMG and the active site of pol ε. At present there is no exact information on the length of exposed leading strand template during coupled synthesis. Current estimates are at least 16 nt, based on a recent structure of pol ε bound to CMG[80]. Importantly, the minimum length required to form a three stacked G4 or i-motif is 15 nt, whereas hairpins could nucleate from even shorter sequences. Very recent super resolution imaging of individual replication forks in human cells have detected G4 structures behind CMG, but not in front of it[81], providing further support for our model that structures form as a consequence of replication.

We have discovered that the replisome can intrinsically resolve stalls induced by hairpin-forming sequences through multiple mechanisms, with pol δ playing a major role. In contrast, stalls induced by quadruplex-forming sequences require the extrinsic support of the accessory helicase Pif1. Our results are in strong agreement with a recent high-throughput primer extension assay that tested the ability of T7 polymerase to extend through all possible 1–6 nt long direct repeats, as well as a large library of hairpin, G4 and i-motif sequences[32]. Synthesis by this model polymerase gradually progressed through hairpins, with more stable hairpins taking longer to resolve, but was terminally stalled at either G4s or i-motifs. It thus seems that quadruplexes are a more robust block to synthesis by many polymerases. In contrast, we found that the two major eukaryotic replicative polymerases exhibit varying intrinsic capacities to synthesise through hairpins. The strand displacement activity of pol δ most likely evolved for the purpose of Okazaki fragment maturation. However, this comes with the added benefit of allowing pol δ to rescue leading strand stalls caused by hairpin-forming sequences.

Replication fork uncoupling leads to exposure of ssDNA on the leading strand template, threatening genetic and epigenetic stability. It is therefore essential to minimise these events. Although pol δ was able to resume synthesis of hairpin-forming repeats on the leading strand, local uncoupling was not completely prevented. Several types of DNA lesions on the leading strand template induce events similar to those we observed here, including inhibition of synthesis and uncoupling of synthesis from unwinding. Interestingly, similar to its ability to synthesise past hairpin-forming sequences, pol δ could also rescue leading strand synthesis past 8-oxoguanine and thymine glycol[82]. In contrast, replication past an abasic site or a CPD could not be carried out by any of the replicative polymerases[5,6]. However, translesion synthesis by pol η could perform synthesis past a CPD[83]. This requirement is very much akin to the role of Pif1 in rescuing replication of quadruplex-forming sequences. Thus, the molecular events that underlie DNA-induced stalling could be mechanistically analogous to those induced by leading strand DNA lesions, exhibiting both intrinsic and extrinsic recovery pathways.

In summary, we have shown that repetitive DNA is an important potential source of endogenous replication stress and have revealed how the eukaryotic replisome is able to cope with difficult-to-replicate sequences. The response of the replisome to certain repetitive sequences is mechanistically similar to events driven by leading strand DNA lesions. We therefore propose that repetitive sequences per se can also induce the replication stress checkpoint response. Thus, our broad knowledge and understanding of the cellular response to replication stress and DNA damaging agents may now be extended to encompass DNA-induced replication stalling.

## Methods

**Cloning**. All replication templates are based on the 9.7 kb pZN3 plasmid[5], in which a new linker was inserted 3 kb downstream from the ARS306 origin, yielding pGC504. Repeats were cloned step-wise using a previously described method for expansion of repeats[37]. Briefly, repeats were first cloned using annealed oligonucleotides. For the first expansion step, annealed duplexes were used as a source of insert. In subsequent steps, each replication template was used both as a source of insert and as a target vector. The use of type IIS restriction enzymes (BsaI and Esp3I) allowed seamless cloning of uninterrupted repeats. Because of the unstable nature of some repeats, we first cloned repeats into a pSMART derivative in which we removed a BsaI site and introduced a new linker. Although this vector has been designed to better support unstable inserts, we found that repeats were overall more stable in the pZN3 backbone. We therefore eliminated two BsaI sites from pGC504, to generate pGC542, and from that point onward cloned all repeats directly into pGC542. To clone repeats in the reverse orientation we replaced the linker in pGC542 so that the PacI and NotI sites were reversed, yielding pGC558. See Table 1 for complete annotation of all plasmids used and generated in this study and Table 2 for a list of oligonucleotides.

**Protein expression and purification**. The expression and purification of most proteins used in this study have been described before[3,36,84–91]. For full details see Table 3. To generate the pol ε P301R expression strain (ySW1), a synthetic gene fragment spanning part of pol2 which contains the desired mutation (ordered as a gBlock, IDT) was cloned using HiFi assembly (New England Biolabs, E2621S) to replace the corresponding WT sequence in plasmid pAJ6, yielding plasmid pSW62. Plasmid pSW62 was linearised with Bsu36I, transformed into yeast strain yAE94[36] and positive transformants were selected for on plates lacking TRP. Integration was confirmed by PCR of genomic DNA as described[87]. WT and mutant pol ε variants were purified as previously described[36] except that yeast cultures were not synchronised.

Pif1 and Pif1 K264A were expressed and purified as described[84] with the following modifications: imidazole concentrations were 0 mM during lysis, 15 mM during washes and 300 mM for elution. The eluate from the HIS pulldown was diluted 1:2 to reduce salt and loaded on a monoS column. Pif1 containing fractions were concentrated and loaded onto a 24 ml Superdex 200 column equilibrated in 0.15 mM NaCl. Pif1 was concentrated with a 30 kDa Amicon to 2 μM.

For Sld2 expression, pGC441 was transformed into BL21 bacteria and grown overnight in a starter culture of 250 ml LB-broth with 100 μg/ml ampicillin and 37 μg/ml chloramphenicol at 37 °C. The next day 20 ml per litre starter was added to 12 litre of LB with ampicillin and chloramphenicol, incubated at 37 °C until OD(600) reached 0.5, then cooled on ice for 20 min, and IPTG was added to 0.2 mM IPTG. Induction took place at 16 °C overnight. Cells were harvested by centrifugation and pellets resuspended in buffer S [25 mM HEPES pH 7.6, 10% glycerol, 0.02% NP-40, 0.1% Tween, 1 mM EDTA, 1 mM DTT] + 0.5 M NaCl and protease inhibitors, incubated with 0.1 mg/ml lysozyme for 20 mins at 4 °C and sonicated for 4 mins (5 s on/5 s off) on ice. The lysate was cleared by centrifugation at $21,000 \times g$, 15 min, 4 °C using a JA-25.50 rotor. The cleared lysate was incubated 1 hour at 4 °C with 2.4 ml of 20% glutathione agarose slurry pre-washed in lysis buffer. The beads were washed extensively with buffer S + 0.5 M NaCl and finally resuspended in 3 ml wash buffer with 200 μg PreScission protease and incubated on rotating wheel 2 h at 4 °C. The eluate was collected, the beads washed with 3 × 1 ml buffer S + 0.5 M NaCl and all fractions were pooled and diluted 1:2 in buffer S without salt. The sample was applied to a HiTrap SP FF 1 ml column equilibrated in buffer S + 250 mM NaCl. After washing with 20 CV of equilibration buffer Sld2 was eluted in 0.5 ml fractions with 100% buffer S + 700 mM NaCl for 10 CV. Sld2-containing fractions were pooled and concentrated with a 10 kDa Amicon to 1.1 μM.

**Preparation of templates for replication assays**. All plasmids were maintained in NEB Stable E. coli cells (New England Biolabs, C3040H) and purified using the HiSpeed Plasmid Maxi kits (Qiagen, 12663) from bacteria grown at 30 °C to minimize loss or rearrangements of unstable inserts. We sometimes observed variability in the overall efficiency of in vitro replication between substrates, presumably due to a contaminant. This variability was eliminated by further purifying templates in batch using PlasmidSelect Xtra resin (VWR, 28-4024-02)

**Table 1 Plasmids.**

| Plasmid | Insert | Backbone | Vector | Vector sites | Insert sites | Derivation of insert | Source |
|---|---|---|---|---|---|---|---|
| Replication templates—empty vectors | | | | | | | |
| pZN3 | | | | | | | Taylor and Yeeles[5] |
| pGC504 | BamHI/PacI/BbvCI/NotI/PstI linker | pZN3 | pZN3 | BamHI PstI | BamHI PstI | Annealed oligo GC497/GC498 | This study |
| pGC542 | pGC504 - BsaI sites removed | pZN3 | pGC504 | | | HiFi mutagenesis with primers GC510/GC511/GC512/GC513 | This study |
| pGC558 | BamHI/NotI/BbvCI/PacI/PstI linker | pZN3 | pGC542 | BamHI PstI | BamHI PstI | Annealed oligos GC520/GC521 | This study |
| pSMART HC-Amp | | | | | | | Lucigen Cat. 40041-2 |
| pGC481 | pSMART HC with SalI NheI linker | pSMART | pSMART HC-Amp | Blunt | Blunt | Annealed oligos GC419/GC420 | This study |
| pGC483 | pSMART HC - BsaI site removed | pSMART | pGC481 | | | Mutagenesis with primers GC421/GC422 | This study |
| Replication templates—Homopolymers | | | | | | | |
| pGC491 | 30xA | pSMART | pGC483 | SalI NheI | SalI NheI | Annealed oligos GC365/GC366 | This study |
| pGC499 | 60xA | pSMART | pGC491 | NotI Esp3I | NotI BsaI | Annealed oligos GC367/GC368 | This study |
| pGC523 | 114xA | pSMART | pGC499 | NotI Esp3I | NotI BsaI | pGC499 | This study |
| pGC526 | 222xA | pSMART | pGC523 | NotI Esp3I | NotI BsaI | pGC523 | This study |
| pGC536 | 222xA | pZN3 | pGC504 | PacI NotI | PacI NotI | pGC526 | This study |
| pGC601 | 222xT | pZN3 | pGC558 | PacI NotI | PacI NotI | pGC536 | This study |
| pGC554 | 10xG | pZN3 | pGC504 | PacI NotI | PacI NotI | Annealed oligos GC516/GC517 | This study |
| pGC606 | 13xG | pZN3 | pGC542 | PacI NotI | PacI NotI | Annealed oligos GC583/GC584 | This study |
| pGC607 | 16xG | pZN3 | pGC542 | PacI NotI | PacI NotI | Annealed oligos GC585/GC586 | This study |
| pGC556 | 20xG | pZN3 | pGC504 | PacI NotI | PacI NotI | Annealed oligos GC518/GC519 | This study |
| pGC543 | 30xG | pZN3 | pGC542 | PacI NotI | PacI NotI | Annealed oligos GC369/GC370 | This study |
| pGC547 | 40xG | pZN3 | pGC543 | NotI Esp3I | | Annealed oligos GC514/GC515 | This study |
| pGC548 | 50xG | pZN3 | pGC547 | NotI Esp3I | | Annealed oligos GC514/GC515 | This study |
| pGC581 | 10xC | pZN3 | pGC558 | PacI NotI | PacI NotI | pGC554 | This study |
| pGC608 | 13xC | pZN3 | pGC558 | PacI NotI | PacI NotI | Annealed oligos GC583/GC584 | This study |
| pGC609 | 16xC | pZN3 | pGC558 | PacI NotI | PacI NotI | Annealed oligos GC585/GC586 | This study |
| pGC582 | 20xC | pZN3 | pGC558 | PacI NotI | PacI NotI | pGC556 | This study |
| pGC583 | 30xC | pZN3 | pGC558 | PacI NotI | PacI NotI | pGC543 | This study |
| pGC584 | 40xC | pZN3 | pGC558 | PacI NotI | PacI NotI | pGC547 | This study |
| pGC585 | 50xC | pZN3 | pGC558 | PacI NotI | PacI NotI | pGC548 | This study |

**Table 1 (continued)**

| Plasmid | Insert | Backbone | Vector | Vector sites | Insert sites | Derivation of insert | Source |
|---|---|---|---|---|---|---|---|
| *Replication templates—dinucleotides* | | | | | | | |
| pGC605 | 6xCG | pZN3 | pGC542 | BamHI PstI | BamHI PstI | Annealed oligos GC581/GC582 | This study |
| pGC604 | 8xCG | pZN3 | pGC542 | BamHI PstI | BamHI PstI | Annealed oligos GC579/GC580 | This study |
| pGC603 | 10xCG | pZN3 | pGC542 | BamHI PstI | BamHI PstI | Annealed oligos GC577/GC578 | This study |
| pGC602 | 12xCG | pZN3 | pGC542 | BamHI PstI | BamHI PstI | Annealed oligos GC575/GC576 | This study |
| pGC566 | 14xCG | pZN3 | pGC542 | BamHI PstI | BamHI PstI | Annealed oligos GC544/GC545 | This study |
| pGC593 | 24xCG | pZN3 | pGC566 | NotI Esp3I | | Annealed oligos GC546/GC547 | This study |
| pSW23 | 24xCG Scramble 1 | pZN3 | pGC542 | PacI NotI | PacI NotI | Annealed oligos SW015/SW016 | This study |
| pSW24 | 24xCG Scramble 2 | pZN3 | pGC542 | PacI NotI | PacI NotI | Annealed oligos SW017/SW018 | This study |
| pGC567 | 14xCA | pZN3 | pGC542 | BamHI PstI | BamHI PstI | Annealed oligos GC548/GC549 | This study |
| pGC594 | 25xCA | pZN3 | pGC567 | NotI Esp3I | | Annealed oligos GC550/GC551 | This study |
| pGC565 | 15xTA | pZN3 | pGC542 | BamHI PstI | BamHI PstI | Annealed oligos GC540/GC541 | This study |
| pGC634 | 25xTG | pZN3 | pGC558 | BamHI PstI | | pGC594 | This study |
| pGC633 | 25xGA | pZN3 | pGC542 | BamHI PstI | BamHI PstI | Annealed oligos GC641/GC642 | This study |
| *Replication templates—trinucleotides* | | | | | | | |
| pGC570 | 11xCTG | pZN3 | pGC542 | BamHI PstI | BamHI PstI | Annealed oligos GC560/GC561 | This study |
| pGC597 | 21xCTG | pZN3 | pGC570 | NotI Esp3I | | Annealed oligos GC562/GC563 | This study |
| pSW01 | 41xCTG | pZN3 | pGC597 | NotI Esp3I | NotI BsaI | pGC597 | This study |
| pSW02 | 81xCTG | pZN3 | pSW01 | NotI Esp3I | NotI BsaI | pSW01 | This study |
| pSW03 | 161xCTG | pZN3 | pSW02 | NotI Esp3I | NotI BsaI | pSW02 | This study |
| pSW08 | 161xCAG | pZN3 | pGC558 | PacI NotI | PacI NotI | pSW03 | This study |
| pGC489 | 11xCGG | pSMART | pGC483 | SalI NheI | SalI NheI | Annealed oligos GC363/364 | This study |
| pGC497 | 21xCGG | pSMART | pGC489 | NotI Esp3I | NotI BsaI | Annealed oligos GC363/364 | This study |
| pGC503 | 41xCGG | pSMART | pGC497 | NotI Esp3I | NotI BsaI | pGC497 | This study |
| pGC541 | 61xCGG | pSMART | pGC503 | pGC503 | NotI BsaI | pGC497 | This study |
| pGC507 | 11xCGG | pZN3 | pGC504 | PacI NotI | PacI NotI | pGC489 | This study |
| pGC610 | 14xCGG | pZN3 | pGC542 | PacI NotI | PacI NotI | Annealed oligos GC587/GC588 | This study |
| pGC611 | 17xCGG | pZN3 | pGC542 | PacI NotI | PacI NotI | Annealed oligos GC589/GC590 | This study |
| pGC515 | 21xCGG | pZN3 | pGC504 | PacI NotI | PacI NotI | pGC497 | This study |
| pGC544 | 41xCGG | pZN3 | pGC542 | PacI NotI | PacI NotI | pGC503 | This study |
| pGC545 | 61xCGG | pZN3 | pGC504 | PacI NotI | PacI NotI | pGC541 | This study |
| pGC549 | 81xCGG | pZN3 | pGC544 | NotI Esp3I | NotI BsaI | pGC544 | This study |

**Table 1 (continued)**

| Plasmid | Insert | Backbone | Vector | Vector sites | Insert sites | Derivation of insert | Source |
|---|---|---|---|---|---|---|---|
| pGC551 | 101xCGG | pZN3 | pGC549 | NotI Esp3I | NotI BsaI | pGC497 | This study |
| pGC552 | 121xCGG | pZN3 | pGC549 | NotI Esp3I | NotI BsaI | pGC544 | This study |
| pGC553 | 161xCGG | pZN3 | pGC549 | NotI Esp3I | NotI BsaI | pGC549 | This study |
| pMdm10 | 21xCGG Scramble 1 | pZN3 | pGC542 | PacI NotI | PacI NotI | Annealed oligos pMDM10F/ pMDM10R | This study |
| pMdm11 | 21xCGG Scramble 2 | pZN3 | pGC542 | PacI NotI | PacI NotI | Annealed oligos pMDM11F/ pMDM11R | This study |
| pGC571 | 11xCCG | pZN3 | pGC558 | PacI NotI | PacI NotI | pGC507 | This study |
| pGC612 | 14xCCG | pZN3 | pGC558 | PacI NotI | PacI NotI | Annealed oligos GC587/ GC588 | This study |
| pGC613 | 17xCCG | pZN3 | pGC558 | PacI NotI | PacI NotI | Annealed oligos GC589/ GC590 | This study |
| pGC572 | 21xCCG | pZN3 | pGC558 | PacI NotI | PacI NotI | pGC515 | This study |
| pGC573 | 41xCCG | pZN3 | pGC558 | PacI NotI | PacI NotI | pGC503 | This study |
| pGC574 | 61xCCG | pZN3 | pGC558 | PacI NotI | PacI NotI | pGC545 | This study |
| pGC487 | 11xGAA | pSMART | pGC483 | SalI NheI | SalI NheI | Annealed oligos GC361/ GC362 | This study |
| pGC495 | 21xGAA | pSMART | pGC487 | NotI Esp3I | NotI BsaI | Annealed oligos GC361/ GC362 | This study |
| pGC522 | 41xGAA | pSMART | pGC495 | NotI Esp3I | NotI BsaI | pGC495 | This study |
| pGC525 | 81xGAA | pSMART | pGC522 | NotI Esp3I | NotI BsaI | pGC522 | This study |
| pGC535 | 161xGAA | pSMART | pGC525 | NotI Esp3I | NotI BsaI | pGC525 | This study |
| pGC538 | 161xGAA | pZN3 | pGC504 | PacI NotI | PacI NotI | pGC535 | This study |
| pGC580 | 161xTTC | pZN3 | pGC558 | PacI NotI | PacI NotI | pGC538 | This study |
| **Protein expression** | | | | | | | |
| pGEX-4T1 Sld2 | GST-Thrombin-Sld2 | pGEX-4T-1 | pGEX-4T-1 | EcoRI XhoI | EcoRI XhoI | Budding yeast Sld2 PCR amplified from genomic DNA | Diffley lab (stock 1680) |
| pGC441 | GST-PreScission-Sld2 | pGEX-6P-1 | pGEX-6P-1 | EcoRI XhoI | EcoRI XhoI | pGEX-4T1 Sld2 | This study |
| pAJ6 | Pol2-Gal1,10-Dpb4-TEV-CBP | pRS304 | pAJ6 | | | | Yeeles et al.[36] |
| pSW62 | Pol2-P301R-Gal1,10-Dpb4-TEV-CBP | pRS304 | pAJ6 | BlpI MscI | BlpI MscI | HiFi cloning of a synthetic gBlock fragment | This study |

**Table 2 Primers.**

| Oligo name | Oligo sequence | Source |
|---|---|---|
| GC361 — 11xGAA F | AAAAGTCGACTTAATTAAGGTCTCAGAAGAAGAAGAAGAAGAAGAAGAAGAAGAAGAAG AGAGACGGCGGCCGCGCTAGCAAAA | This study |
| GC362 — 11xGAA R | TTTTGCTAGCGCGGCCGCCGTCTCTCTTCTTCTTCTTCTTCTTCTTCTTCTTCTTCTGAGAC CTTAATTAAGTCGACTTTT | This study |
| GC363 — 11xCGG F | AAAAGTCGACTTAATTAAGGTCTCACGGCGGCGGCGGCGGCGGCGGCGGCGGCGGCAGA GACGGCGGCCGCGCTAGCAAAA | This study |
| GC364 — 11xCGG R | TTTTGCTAGCGCGGCCGCCGTCTCTGCCGCCGCCGCCGCCGCCGCCGCCGCCGCCGTGAGAC CTTAATTAAGTCGACTTTT | This study |
| GC365 — 30xA F | AAAAGTCGACTTAATTAAGGTCTCAAAAAAAAAAAAAAAAAAAAAAAAAAAAAAGAGACGGCG GCCGCGCTAGCAAAA | This study |
| GC366 — 30xA R | TTTTGCTAGCGCGGCCGCCGTCTCTTTTTTTTTTTTTTTTTTTTTTTTTTTTTTGAGACCTTAATTAAGT CGACTTTT | This study |
| GC367 — 36xA F | AAAAGTCGACTTAATTAAGGTCTCAAAAAAAAAAAAAAAAAAAAAAAAAAAAAAAAAAAAGAGA CGGCGGCCGCGCTAGCAAAA | This study |
| GC368 — 36xA R | TTTTGCTAGCGCGGCCGCCGTCTCTTTTTTTTTTTTTTTTTTTTTTTTTTTTTTTTTTTTGAGACCTTAATT AAGTCGACTTTT | This study |
| GC369 — 30xG F | AAAAGTCGACTTAATTAAGGTCTCGGGGGGGGGGGGGGGGGGGGGGGGGGGGGGGAGACGGCGGCC GCGCTAGCAAAA | This study |
| GC370 — 30xG R | TTTTGCTAGCGCGGCCGCCGTCTCCCCCCCCCCCCCCCCCCCCCCCCCCCCCCCGAGACCTTAATTAAG TCGACTTTT | This study |
| GC419 - pSMART linker SalI NheI | GTCGACCACCAACACAACGCTATCGGGCGATTCTATGCTAGC | This study |
| GC420 - pSMART linker SalI NheI | GCTAGCATAGAATCGCCCGATAGCGTTGTGTTGGTGGTCGAC | This study |
| GC421 - Mutate BsaI site in pSMART F | CCCGCGGTATCATTGCAGCACT | This study |
| GC422 - Mutate BsaI site in pSMART R | AGCCACGCTCACCGGCTCCA | This study |
| GC497 - BamHI/PacI/NotI/PstI linker F | GATCCTTAATTAACCTCAGCTTGACCATGACTCGACTGCAATCGCCCTCAGCGCGGCCGCCTGCA | This study |
| GC498 - BamHI/PacI/NotI/PstI linker R | GGCGGCCGCGCTGAGGGCGATTGCAGTCGAGTCATGGTCAAGCTGAGGTTAATTAAG | This study |
| GC510 - HiFi mutate BsaI sites in pZN3 F1 | GAGCGTGGATCGCGCGGTATCATTGCAGCACTGG | This study |
| GC511 - HiFi mutate BsaI sites in pZN3 R1 | AAGACATGCTGACCAAGTGGCAAAATCCAACG | This study |
| GC512 - HiFi mutate BsaI sites in pZN3 F2 | CCACTTGGTCAGCATGTCTTGCTTTGGTG | This study |
| GC513 - HiFi mutate BsaI sites in pZN3 R2 | ATACCGCGCGATCCACGCTCACCGGCTCCAGATT | This study |
| GC514 — 16xG F | GGGGGGGGGGGGGGGGGAGACGGC | This study |
| GC515 — 16xG R | GGCCGCCGTCTCCCCCCCCCCCCCC | This study |
| GC516 — 10xG F | TAAGGTCTCGGGGGGGGGGGGAGACGGC | This study |
| GC517 — 10xG R | GGCCGCCGTCTCCCCCCCCCCCCGAGACCTTAAT | This study |
| GC518 — 20xG F | TAAGGTCTCGGGGGGGGGGGGGGGGGGGGGGAGACGGC | This study |
| GC519 — 20xG R | GGCCGCCGTCTCCCCCCCCCCCCCCCCCCCCCCGAGACCTTAAT | This study |
| GC520 - Reverse linker BamHI/NotI/PacI/PstI F | GATCCGCGGCCGCCCTCAGCTTGACCATGACTCGACTGCAATCGCCCTCAGCTTAATTAACTGCA | This study |
| GC521 - Reverse linker BamHI/NotI/PacI/PstI R | GTTAATTAAGCTGAGGGCGATTGCAGTCGAGTCATGGTCAAGCTGAGGGCGGCCGCG | This study |
| GC540 — 15xTA F | GATCCTTAATTAAGGTCTCTATATATATATATATATATATATATATATAGAGACGGCGGCCGCCTGCA | This study |
| GC541 — 15xTA R | GGCGGCCGCCGTCTCTATATATATATATATATATATATATATATAGAGACCTTAATTAAG | This study |
| GC544 — 14xCG F | GATCCTTAATTAAGGTCTCGCGCGCGCGCGCGCGCGCGCGCGCGCGAGACGGCGGCCGCCTGCA | This study |
| GC545 — 14xCG R | GGCGGCCGCCGTCTCGCGCGCGCGCGCGCGCGCGCGCGCGCGGAGACCTTAATTAAG | This study |
| GC546 - CG Expand F | CGCGCGCGCGCGCGCGCGCGCGCGCGAGACGGC | This study |
| GC547 - CG Expand R | GGCCGCCGTCTCGCGCGCGCGCGCGCGCGCGCG | This study |
| GC548 — 14xCA Clone F | GATCCTTAATTAAGGTCTCACACACACACACACACACACACACACACGAGACGGCGGCCGCCTGCA | This study |
| GC549 — 14xCA Clone R | GGCGGCCGCCGTCTCGTGTGTGTGTGTGTGTGTGTGTGTGTGTGAGACCTTAATTAAG | This study |
| GC550 - CA Expand F | CACACACACACACACACACACACACACGAGACGGC | This study |
| GC551 - CA Expand R | GGCCGCCGTCTCGTGTGTGTGTGTGTGTGTGTGTGT | This study |
| GC560 — 11xCTG F | GATCCTTAATTAAGGTCTCACTGCTGCTGCTGCTGCTGCTGCTGCTGCTGCTGCAGAGACGGCGGCCGCCTGCA | This study |
| GC561 — 11xCTG R | GGCGGCCGCCGTCTCTGCAGCAGCAGCAGCAGCAGCAGCAGCAGCAGCAGTGAGACCTTAATTAAG | This study |
| GC562 - CTG Expand F | CTGCTGCTGCTGCTGCTGCTGCTGCTGCAGAGACGGC | This study |
| GC563 - CTG Expand R | GGCCGCCGTCTCTGCAGCAGCAGCAGCAGCAGCAGCAGCA | This study |
| GC575 — 12xCG F | GATCCTTAATTAAGGTCTCGCGCGCGCGCGCGCGCGCGCGCGAGACGGCGGCCGCCTGCA | This study |
| GC576 — 12xCG R | GGCGGCCGCCGTCTCGCGCGCGCGCGCGCGCGCGCGCGGAGACCTTAATTAAG | This study |
| GC577 — 10xCG F | GATCCTTAATTAAGGTCTCGCGCGCGCGCGCGCGCGCGAGACGGCGGCCGCCTGCA | This study |
| GC578 — 10xCG R | GGCGGCCGCCGTCTCGCGCGCGCGCGCGCGCGCGGAGACCTTAATTAAG | This study |
| GC579 — 8xCG F | GATCCTTAATTAAGGTCTCGCGCGCGCGCGCGCGAGACGGCGGCCGCCTGCA | This study |
| GC580 — 8xCG R | GGCGGCCGCCGTCTCGCGCGCGCGCGCGCGGAGACCTTAATTAAG | This study |
| GC581 — 6xCG F | GATCCTTAATTAAGGTCTCGCGCGCGCGCGAGACGGCGGCCGCCTGCA | This study |
| GC582 — 6xCG R | GGCGGCCGCCGTCTCGCGCGCGCGCGGAGACCTTAATTAAG | This study |
| GC583 — 13xG F | TAAGGTCTCGGGGGGGGGGGGGGGAGACGGC | This study |
| GC584 — 13xG R | GGCCGCCGTCTCCCCCCCCCCCCCCCGAGACCTTAAT | This study |
| GC585 — 16xG F | TAAGGTCTCGGGGGGGGGGGGGGGGGGAGACGGC | This study |
| GC586 — 16xG R | GGCCGCCGTCTCCCCCCCCCCCCCCCCCCGAGACCTTAAT | This study |

**Table 2 (continued)**

| Oligo name | Oligo sequence | Source |
|---|---|---|
| GC587 — 14xCGG F | TAAGGTCTCACGGCGGCGGCGGCGGCGGCGGCGGCGGCGGCGGCGGCGGCAGAGACGGC | This study |
| GC588 — 14xCGG R | GGCCGCCGTCTCTGCCGCCGCCGCCGCCGCCGCCGCCGCCGCCGCCGCCGCCGTGAGACCTTAAT | This study |
| GC589 — 17xCGG F | TAAGGTCTCACGGCGGCGGCGGCGGCGGCGGCGGCGGCGGCGGCGGCGGCGGCGGCGGCGCAGAGACGGC | This study |
| GC590 — 17xCGG R | GGCCGCCGTCTCTGCCGCCGCCGCCGCCGCCGCCGCCGCCGCCGCCGCCGCCGCCGCCGTGAGACCTTAAT | This study |
| GC641 — 25xGA F | TAAGGTCTCGAGAGAGAGAGAGAGAGAGAGAGAGAGAGAGAGAGAGAGAGAGAGAGACAGAGACGGC | This study |
| GC642 — 25xGA R | GGCCGCCGTCTCTGTCTCTCTCTCTCTCTCTCTCTCTCTCTCTCTCTCTCTCTCTCTCGAGACCTTAAT | This study |
| pMdm10F — 21xCGG Scramble 1 | TAAGGTCTCGGGCGGCGGGCGCGCCGCGGGGGGGCGCGGGGCCCGGCGGGGGCGCGGGGCGGCGGGCGCGCGAGACGGC | This study |
| pMdm10R — 21xCGG Scramble 1 | GGCCGCCGTCTCGCGCGCCCGCCGCCCCGCGCCCCCGCCGGGCCCCGCGCCCCCCCGCGGCGCGCCCGCCGCCCGAGACCTTAAT | This study |
| pMdm11F — 21xCGG Scramble 2 | TAAGGTCTCGGCGGCGGCGGGCGGGGGGGCCGGGGGCCGCCCGCGGGCGGGGCCGGGGCGGCGGCGGGCGGGAGACGGC | This study |
| pMdm11R — 21xCGG Scramble 2 | GGCCGCCGTCTCCCGCCCGCCGCCGCCCCGGCCCCGCCCGCGGGCGGCCCCCGGCCCCCCCCGCCCGCCGCCGCCGAGACCTTAAT | This study |
| SW015 — 24xCG Scramble 1F | TAAGGTCTCCGGCCCCGCGCCGGGCCGCGCCGCGCGGGCGCCCCGCGCGCCGGGGGGGAGACGGC | This study |
| SW016 — 24xCG Scramble 1R | GGCCGCCGTCTCCCCCGGCGCGCGGGGCGCCCGCGCGGCGCGGCCCCGGCGCGGGGCCGGAGACCTTAAT | This study |
| SW017 — 24xCG Scramble 2F | TAAGGTCTCGCCCGCGCGCCCGGCCCGGCCCGCCCGGGGGCCCGCCCGCGGCCGCGGGGCGGCGGGAGACGGC | This study |
| SW018 — 24xCG Scramble 2R | GGCCGCCGTCTCCCGCCGCCCCGCGGGCCGCGGGCGGGCCCCGGGCCGGGCCGCGCGGGCGAGACCTTAAT | This study |
| Repriming 265nt | CTGGTTTCCGCCGT | Taylor and Yeeles[5] |
| Repriming 265nt scrambled | GTTCGCCTTGCCGT | Taylor and Yeeles[5] |
| AflIII undamaged | 5′-phos-TCAGCACTTAAGTCC | Taylor and Yeeles, Mol Cell[5] |
| AflIII CPD | 5′-phos-TCAGCAC-/CPD/-AAGTCC | Taylor and Yeeles[5] |
| AflIII competitor | GTTGGTACTGCGCTGAGGACTTAAGTGCTGAGGTTGGTACTGCG | Taylor and Yeeles[5] |

as follows: Plasmid DNA was diluted fivefold in 3 M ammonium sulfate, added to 300 µl bead slurry pre-washed with 2.3 M ammonium sulfate and incubated for 30 min rotating. The beads were washed four times with 1.9 M ammonium sulfate. Supercoiled plasmid was eluted with 2 × 1 ml of 1.5 M ammonium sulfate. Both fractions were pooled and dialysed for 3 h and ON against 1 L of 0.1 × TE buffer in the dark using in a D-Tube Dialyzer Mini, MWCO 6-8 kDa (Merck, 71504). The dialysed DNA sample was concentrated to 400 µl with a 100 kDa Amicon, precipitated with 1 ml 100% ethanol and 90 mM NaCl and resuspended in TE. All steps except for the ethanol precipitation were performed at RT. All ammonium sulfate buffers contained 100 mM Tris HCl pH 7.5 and 10 mM EDTA.

**In vitro replication assays**. For MCM loading and phosphorylation, 3 nM plasmid DNA template was incubated with 5 mM ATP, 75 nM Cdt1/Mcm2-7, 45 nM Cdc6, 20 nM ORC, 50 nM DDK in 25 mM HEPES-KOH (pH 7.6), 100 mM potassium glutamate, 0.01% NP-40-S, 1 mM DTT, 10 mM Mg(OAc)2, 0.1 mg/ml BSA (1× reaction buffer) and 80 mM KCl at 24 °C for 10 min. For replication of linearised templates 0.6 U/µl AhdI was added during MCM loading. For truncated templates in Fig. 4C, D, 2 U/µl EcoRV was also added. Loading was stopped by adding 120 nM S-CDK for further 5 min. The loaded reaction was diluted in 1× reaction buffer so that the final dilution in the replication reaction was sixfold. To start replication, the following components were added to the reaction: 200 µM CTP, GTP, UTP, 30 µM dATP, dCTP, dGTP, dTTP, 33 nM α-[33 P]-dATP, 5 mM ATP, 10 nM S-CDK, 30 nM Dpb11, 100 nM GINS, 40 nM Cdc45, 20 nM Pol ε, 10 nM Mcm10, 40 nM RPA, 20 nM Csm3/Tof1, 20 nM Mrc1, 30 nM RFC, 40 nM PCNA, 10 nM TopoI (for circular reactions), 40 nM Pol α, 5 nM Pol δ (where indicated), 20 nM Sld3/7, 20 nM Sld2, and the mix was incubated at 30 °C for the indicated time. For samples loaded on denaturing gels, 0.5 U/µl SmaI was added 5 min before the end of the reaction, which eliminates product length heterogeneity which stems from variable initiation sites[5]. Reactions were stopped with 100 mM EDTA. For pulse-chase experiments, unlabelled deoxyribonucleotide concentrations were adjusted during the pulse to 30 µM dCTP, dTTP, dGTP and 2.5 µM dATP, or 7.5 µM dATP for experiments without RFC/PCNA. After a 10 min pulse, the chase was performed by adding 200 µM unlabelled dATP, or for Fig. 5B 400 µM of dATP alone or dCTP, dTTP, dGTP and dATP, as indicated. Reactions were stopped at the indicated time point by addition of EDTA to 100 mM. For repriming experiments: oligonucleotides were added to 60 nM (molecules) before starting the replication reaction.

**Post-reaction sample processing**. For samples to be analysed on denaturing gels, alkaline loading dye (0.5 M NaOH, 10% sucrose, xylene cyanol in water) was added at 1/10 volume. Samples were loaded in denaturing 0.8% agarose gels run at 32 V overnight in 30 mM NaOH, 2 mM EDTA.

For reactions to be loaded on native gels, SDS (to 0.1%) and proteinase K (1/100 volumes) were added and incubated at 37 °C for 20 min. The sample volume was increased to 25 µl with TE and DNA was extracted with phenol:chloroform:isoamyl alcohol 25:24:1 (Sigma-Aldrich, P2069). The extracted sample was mixed with 5× Invitrogen™ Novex™ High-Density TBE Sample Buffer and loaded on a 1% agarose/TAE gel.

All gels were dried onto filter paper, exposed to a 20 × 25 cm Storage Phosphor Screen (GE Healthcare, BAS-IP MS 2025) and scanned with a Typhoon Scanner (Cytiva). Image analysis was carried out with ImageJ v1.51.

**Substrate preparation for helicase assays**. Complementary oligonucleotides containing a 5′ overhang were resuspended to 10 µM in 10 mM Tris pH-8.0. One oligo was labelled in a reaction containing 5 pmol of DNA, 1X PNK buffer, 1U of PNK enzyme (NEB, M0201S), and γ-P32-ATP (0.03 mCi). The reaction was incubated for 30 min at 37 °C and subsequently heat inactivated for 20 min at 65 °C. Excess γ-P32-ATP was then cleared using a G50 column (GE healthcare, 2753002) and volume adjusted to 100 µl (=50 nM). To generate duplex DNA 1 pmol of labelled oligo was mixed with 1.5 pmol of unlabelled oligo and incubated at 90 °C for 5 min in a thermal cycler. The mix was then gradually cooled down to room temperature over 2 h. Duplex DNA was stored at −20 °C.

**Helicase assays**. Helicase assays were carried out using 0.5 nM γ-P32-ATP labelled duplex with a 5′ overhang in buffer containing 25 mM Hepes 7.6, 2 mM MgOAc, 0.1 mg/ml BSA and 2 mM ATP. Reactions were assembled on ice, equilibrated to room temperature and the respective helicases (Pif1 or Dna2) added to 50 nM final concentration. Reactions were incubated for 30 min at 30 °C and samples collected at different time points (5, 10 and 20 min). Reactions were stopped by addition of 0.5% SDS and 200 mM EDTA. The samples were supplemented with Novex Hi-Density TBE Sample buffer (ThermoFisher Scientific, LC6678) and analysed on 10% Novex TBE gels (ThermoFisher Scientific, EC62755BOX) at 150 V for 1 hour in 1× TBE. Gels were dried onto filter paper, autoradiographed with phosphoscreens imaging plates (Fujifilm) and developed on a Typhoon phophorimager (GE Healthcare).

**Table 3 Protein purification.**

| Protein | Strain | Source | Purified as in |
|---|---|---|---|
| Proteins expressed in budding yeast | | | |
| ORC | ySDORC | Frigola et al.[86] | Frigola et al.[86] |
| Mcm2-7/Cdt1 | yAM33 | Coster et al.[87] | Coster et al.[87] |
| DDK | ySDK8 | On et al.[88] | On et al.[88] |
| CDK | yAE88 | Hill et al.[91] | Yeeles et al.[36] |
| Csm3/Tof1 | yAE48 | Yeeles et al.[3] | Yeeles et al.[3] |
| Pol epsilon | yAJ2 | Yeeles et al.[36] | Yeeles et al.[36] |
| Pol epsilon Exo- (pol2-D290A/ E292A) | yAE99 | Goswami et al.[90] | Yeeles et al.[36] |
| Pol epsilon P301R (pol2-P301R) | ySW1 | This study | Yeeles et al.[36] |
| Dpb11 | yJY26 | Yeeles et al.[36] | Yeeles et al.[36] |
| Cdc45 | yJY13 | Yeeles et al.[36] | Yeeles et al.[36] |
| Mrc1 | yAE71 | Hill et al.[91] | Baretić et al.[85] |
| RPA | yAE31 | Yeeles et al.[36] | Yeeles et al.[36] |
| RFC | yAE41 | Yeeles et al.[36] | Yeeles et al.[36] |
| Pol delta | yAE34 | Yeeles et al.[3] | Yeeles et al.[3] |
| Pol alpha | yAE95 | Hill et al.[91] | Yeeles et al.[36] |
| Sld3/Sld7 | yTD6 | Yeeles et al.[36] | Yeeles et al.[36] |
| Dna2 | yFV43 | Deegan et al.[84] | Deegan et al.[84] |
| Pif1 | pTDK10 | Deegan et al.[84] | Deegan et al.[84] |
| Pif1 K264A | pTDK24 | Deegan et al.[84] | Deegan et al.[84] |
| TopoI | yAE42 | Yeeles et al.[3] | Yeeles et al.[3] |

| Protein | Plasmid | Source | Purified as in |
|---|---|---|---|
| Proteins expressed in bacteria | | | |
| Cdc6 | pAM3 | Frigola et al.[86] | Frigola et al.[86] |
| GINS | pJFDJ5 | Yeeles et al.[36] | Yeeles et al.[36] |
| Mcm10 | pMD132 | Douglas & Diffley[89] | Douglas & Diffley[89] |
| PCNA | pJY19 | Yeeles et al.[3] | Yeeles et al.[3] |
| Sld2 | pGC441 | This study | Detailed in "Methods" section |

**CPD substrate**. Preparation of a substrate containing site-specific DNA damage (CPD) was prepared as previously described[5] with several modifications. An oligonucleotide containing a CPD (AflII CPD, HPLC-purified; TriLink Biotechnology) was synthesised and stored in 10 mM Tris-Hcl (pH 8.0), 1 mM EDTA at −20 °C. To introduce the oligo into the plasmid of interest (pGC504), 4 × 200 µg of the relevant plasmid was cut with 15 µl (150U) of Nt.BbvCI (NEB, R0632) in a 200 µl final volume reaction at 37 °C for 3 h. The reaction was stopped by adding 50 mM EDTA. Following digestion, competitor oligonucleotide (AflII competitor, IDT) was added to 1000-fold molar excess over plasmid concentration (27 µl from 1 mM Stock). The mix was incubated at 50 °C for 20 min, then transferred to 37 °C and SDS added to 0.1%. After 5 min, 1/100 volumes of proteinase K (New England Biolabs P8107S) was added and incubated at 37 °C for a further 15 min. All tubes were then pooled and the gapped plasmid purified. Excess oligo was separated from gapped plasmid using High prep PCR magnetic beads (Magbio, AC-60050) with a ratio of 1.8 µl of bead slurry/µl of sample and binding for 30 min at room temperature. Bound fractions were washed 3 times with a mix containing 70% EtOH and 0.02% NP-40 and then eluted in 1X TE. DNA was pooled and concentration measured. This step usually yielded around 60% of input material.

One hundred micrograms of gapped plasmid was collected per oligonucleotide ligation. Complementary oligonucleotide containing a CPD (AflII CPD) was added at a 20-fold molar excess and incubated at 50 °C for 15 min before gradually letting it cool down to room temperature. One hundred micrograms of DNA was ligated in 1× T4 DNA ligase buffer (NEB: B0202S) and T4 ligase (100U/µg) (NEB: M0202M) plus 2 mM Mg(OAc)2 overnight at 16 °C in the dark. The following day, SDS (to 0.1%) and proteinase K (1/100 volumes) were added and incubated at 37 °C for 20 min. The ligated plasmid was then subjected to CsCl gradients as in ref. 5 to specifically purify fully ligated supercoiled substrates. Following the CsCl gradient DNA was dialysed against two changes of 2 L TE over 16 h total in a D-Tube Dialyzer Midi, MWCO 6–8 kDa (Merck 71507) at 4 °C in the dark to remove all traces of CsCl. The DNA was collected and subjected to ethanol precipitation using 0.3 M NaCl + 2.8 volumes ice cold 100% ethanol in dry ice. The pellet was harvested, washed with room temperature 70% ethanol, harvested, air-dried and resuspended in 50 µl TE. As a control, the exact same procedure was also carried out with an undamaged oligo (AflII undamaged) and the resulting template

replicated in the same manner as the parental template, indicating that the observed stalling was induced by the CPD and not due to the process itself.

**Analytical digestion of substrates**. Substrates of interest were subjected to enzymatic digestion to verify the length of the repetitive sequence. Briefly, 100 ng of plasmid was digested with 0.5U of NotI (NEB, R0189L) and PacI (NEB, R0547L) in 1× Cutsmart buffer (NEB, B7204S) at 37 °C for 30 min. Reactions were stopped by adding 50 mM EDTA. The samples were then supplemented with Novex Hi-Density TBE Sample buffer (ThermoFisher Scientific, LC6678) and analysed on 10% Novex TBE gels (ThermoFisher Scientific, EC62755BOX) at 150 V for 1 h in 1× TBE. Gels were then stained with SYBR™ Gold Nucleic Acid Gel Stain (Invitrogen, S11494) for 20 min at room temperature in the dark and imaged on a Typhoon phophorimager (GE Healthcare).

**Statistics and reproducibility**. All experiments have been repeated with similar results at least three times. All experiments with quantification (Fig. 5A and Suppmentary Fig. 2a) were repeated five times.

**Reporting summary**. Further information on research design is available in the Nature Research Reporting Summary linked to this article.

## Data availability

All data generated or analysed during this study are included in this published article (and its supplementary information files). Source data are provided with this paper.

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

## Acknowledgements
This work was funded by a Wellcome Trust and Royal Society Sir Henry Dale fellowship (210470/Z/18/Z) as well as internal funding from the Institute of Cancer Research. We would like to thank Laurence Pearl, Antony Oliver and Charlotte Fisher (Genome Damage and Stability Centre, University of Sussex, UK) for providing unpublished reagents. We would like to thank Max Douglas for reagents, experimental support and critical reading of the manuscript. We would also like to thank Jonathon Pines, Wojciech Niedzwiedz, Sebastian Guettler, Marco Di Antonio, Christian Zierhut and Allison McClure for critical reading of the manuscript.

## Author contributions
Methodology and Investigation, C.S.C.D, M.D.M, S.W and G.C; Conceptualisation, supervision, writing and funding acquisition, G.C.

## Competing interests
The authors declare no competing interests.
