## [Peer Review File · Nature Communications]

The Mechanism of Replication Stalling and Recovery within Repetitive DNAREVIEWER COMMENTS

Reviewer #1 (Remarks to the Author):

This is an interesting study that shows stalling of the reconstituted yeast replication fork at structure-prone (CGG)_n and (CG)_n repeats. Previously, (CGG)_n repeats were shown to stall purified DNA polymerases and cause replication fork stalling in vivo in various experimental systems ranging from bacteria to human cells. The observation that (CGG)_n repeats cause fork stalling in vitro is an important addition to this earlier literature, as it allows the authors to rule out additional factors potentially contributing to stalling, such as repeat-binding proteins or transcription through the repeat. Admittedly, however, fork stalling observed here for fairly long (CGG)_n repeats was quite modest, as compared to profound replication stalling caused much shorter (CGG)_n repeats in yeast plasmids.

Other structure-forming repeats studied didn't cause detectable fork stalling. Given the differences between the strength of fork stalling in vivo and in vitro for (CGG)_n repeats, this is my not be surprising for (CAG)_n repeats, which cause only a modest fork stalling in vivo. Similarly, only a short (AT)₁₄ repeat was studied here, which didn't cause significant problems for replication in yeast (Kaushal et al., 2019).

The lack of fork stalling in vitro at (GAA)_n repeats is more troublesome. There is overwhelming evidence that these repeats cause fork stalling in bacteria, yeast and human cells. The authors are aware of this literature and are trying to come up with an explanation, primarily concentrating on the differences in the pattern of fork stalling between different experimental systems. An alternative explanation, however, is that the authors' negative result may simply reflect the differences between their in vitro system and replication machinery in vivo. Those may include the stoichiometry of the fork components, nucleotide pool composition, ionic conditions, etc.

To sum up, the positive results with GC-rich repeats are impressive, while negative results with other repeats are not convincing at best.

Additional comments:

- (1) Lines 180-182 mention that the stall is saturated at CGG₆₁ referencing to Figure 1D. At least by looking at the figure, it seems like CGG₈₁ and CGG₁₆₁ cause stronger stalls than CGG₆₁. Thus, quantification is a must.
- (2) The fact that a lot of experiments in the supplement are done in the absence of Pol δ begs a question: could Pol δ contribute to fork stalling as a whole?
- (3) Only repeats forming the most stable alternative DNA structures cause fork stalling in this study. Could it be that in the author's conditions disfavor formation of other secondary structures?
- (4) Figure 5A: P301R rescuing the stall is clear, but I am not so sure about the Exo- condition. The background in lanes 5-6 and especially 11-12 seems much lower, as well as the amount leading strands. Quantification of these data is needed.
- (5) Lines 344-345 and Fig. 5B lane 14: It looks like addition of dA per se rescues the stall at CGG₆₁ in the presence of Pol δ . This fact is not mentioned or discussed in the MS.
- (6) Section starting at 353: Pif1 promotes replication through G4. Elsewhere in the paper, the authors suggest that the CGG repeat is forming a hairpin rather than G4-DNA. Yet addition of Pif1 rescues the stall caused by the CGG repeat (Fig. 6A and 6B). Could this observation indicate that CGG are forming G4 as well? Interestingly, knocking down Pif1 in yeast did not cause an increase in fork stalling of CGG repeats in (Anand et al., 2012).
- (7) Lines 358-361: RPA-binding also counteracts formations of other structures, as it was shown to be able to melt both intra- and intermolecular triplexes at concentrations of around 25 nM (Wu et al., 2008). The effect of RPA levels was not, in fact, studied for "all tested sequences" as stated in line 359: it was not done for GAA and CTG repeats (Figure S7 B and C).

Reviewer #2 (Remarks to the Author):

The article from Coster and colleagues: "The mechanism of replication stalling and recovery within repetitive DNA" uses an innovative research strategy to address the highly significant research question. They use an *in vitro* system that reconstitutes DNA replication and monitor changes of replication progression at mono-, di-, and tri-nucleotide repeats. They observed that hairpins as well as quadruplexes cause replication stalling if the structure is located on the leading strand template for DNA replication. They observed that the kind of stall as well as the recovery differs between kind of folded structures (hairpins vs. quadruplexes).

The manuscript is very well and clear written. In particular the introduction, which is very lengthy, is very clear and explains the working hypothesis and the relevance of their story clear. Although the manuscript is very good, I have a few major points that I have listed below:

1. It seems that in the replication assay (e.g. Fig 1) always the right strand is synthesized better (stronger) in comparison to the left? The left strand gets stronger in the pol delta strain (Fig1C). Is this due to specific structure formation or strand preference? Why does this biased of strand disappear after the addition of CPD (Fig 1D)?
2. What portion of the template is folded into the secondary structure?
3. In addition it has to be demonstrated that indeed a hairpin or a G4 form. CGG also can fold into G4, less stable than G repeats but indeed a G4. Is the effect really due to hairpin formation or does it also differ in dependency of the stability of the G4. I would like to see other G4 sequences (GG, GGG and GGGG harboring G4). In addition Shape, CD, melting, and other experiments are required to show robust formation of the structure and to estimate stability and conformation status.
4. Are all assays been performed under physiological salt conditions? Are the reaction time, pausing differ in response to salt changes? It has been known for example that salt concentration and the type of salt alters the conformation of G4 and other structures. E.g. Li reduces G4 potential whereas K⁺ enhances and stabilizes G4s
5. They demonstrated beautifully that CTG and GAA repeats do not stall replication, only if replication fork fidelity is reduced (dNTP levels, Pol delta..) can they discuss changes observed in repeat expansion patients to intrinsic changes?
6. They elucidated that Pol delta overtime unfold hairpins at repeats. Can they be more specific? What is the timing? Is this relevant *in vivo*?
7. They observed that PCNA assist directly in replication progression. Is this a direct role of PCNA that participate in structure resolution?
8. Can they exclude that G4 on the lagging strand will cause replication pausing? Or is the system not specific for lagging strand changes? It would be nice to discuss literature on structure on the lagging strand in more detail and discuss the power and the drawback of the method more clearly.
9. Can other helicases, in addition to Pif1 (DNA2 is not an unwinding helicase *in vitro*) also support replication progression: e.g. BLM, WRN, FANCD1?

Reviewer #3 (Remarks to the Author):

The authors have systematically studied how repetitive sequences affect leading strand synthesis in a reconstituted *in vitro* replication system. Earlier studies have demonstrated that both DNA polymerase epsilon and DNA polymerase delta have difficulties bypassing repetitive DNA sequences that can form secondary structures. Here the authors demonstrate that such repetitive sequences impede DNA synthesis, even immediately after the template strand passed through the replicative helicase and should thus be relatively free of secondary structures. Furthermore, the authors suggest that the

resulting uncoupling between Pol epsilon and CMG may induce check-point activation similar to when DNA lesions are present in the leading strand. This remains to be shown. Finally the authors demonstrate that Pol delta can assist Pol epsilon in bypassing the secondary structure, and that Pif1 is very efficient in suppressing the stalling by presumably resolving the secondary structure of the DNA. A more succinct discussion would improve the manuscript.

We thank all of the reviewers for taking the time to carefully consider our manuscript and for their valuable feedback. We have performed multiple new experiments and have amended the manuscript text and figures to address all of the comments raised. We believe that the manuscript is much better now and provide a point-by-point response below.

Reviewer #1 (Remarks to the Author):

This is an interesting study that shows stalling of the reconstituted yeast replication fork at structure-prone (CGG)_n and (CG)_n repeats. Previously, (CGG)_n repeats were shown to stall purified DNA polymerases and cause replication fork stalling in vivo in various experimental systems ranging from bacteria to human cells. The observation that (CGG)_n repeats cause fork stalling in vitro is an important addition to this earlier literature, as it allows the authors to rule out additional factors potentially contributing to stalling, such as repeat-binding proteins or transcription through the repeat. Admittedly, however, fork stalling observed here for fairly long (CGG)_n repeats was quite modest, as compared to profound replication stalling caused much shorter (CGG)_n repeats in yeast plasmids.

We thank the reviewer for highlighting how our work provides evidence that the DNA template itself causes fork stalling rather than an indirect effect. The reviewer notes that our results differ from those seen in vivo. While we believe it is important to compare results obtained from in vivo and in vitro studies, we do not think that they must be identical. (CGG)_n repeats cause pronounced stalling in budding yeast and this occurs in both orientations (PMID: 12556494), whereas we observe more mild stalling and only when (CGG)_n is on the leading strand template. We believe that this may reflect additional pathways that are present in vivo and speculate that transcription-replication conflicts may underlie part of the effect seen in budding yeast. We note that studies of cells derived from Fragile X syndrome patients with very long (CGG)_n repeats (n=450) did not reveal significant replication stalling (PMID: 24289922). If the DNA template itself causes stalling, one would expect these very long repeats to induce significant stalling. The discrepancy between observations made in yeast and human cells, and our in vitro work, may either reflect different mechanisms of stalling, or perhaps different efficiency/pathways of resolving stalls.

Other structure-forming repeats studied didn't cause detectable fork stalling. Given the differences between the strength of fork stalling in vivo and in vitro for (CGG)_n repeats, this is my not be surprising for (CAG)_n repeats, which cause only a modest fork stalling in vivo. Similarly, only a short (AT)₁₄ repeat was studied here, which didn't cause significant problems for replication in yeast (Kaushal et al., 2019).

Indeed, most of the other repeats we have tested did not cause significant stalling. The reviewer is absolutely correct – the length of (AT)_n repeats we have tested was not found to cause fragility or inhibit synthesis in previous studies. Despite much effort, our attempts at cloning longer (AT)_n repeats were not successful. We can therefore not comment on the ability of longer (AT)_n repeats to affect replication in vitro. To reflect this, we have added the following text to the manuscript in page 9 after we describe Figure 2A:

“Long (AT)_n repeats (n=34) have been shown to interfere with replication and cause chromosome fragility in budding yeast (Kaushal et al, 2019). Despite much effort, we were not able to generate (AT)_n repeats longer than 15 units, leaving open the question of whether long (AT)_n repeats can stall the replisome in vitro”.

The lack of fork stalling in vitro at (GAA)_n repeats is more troublesome. There is overwhelming evidence that these repeats cause fork stalling in bacteria, yeast and human cells. The authors are aware of this literature and are trying to come up with an explanation, primarily concentrating on the differences in the pattern of fork stalling between different experimental systems. An alternative explanation, however, is that the authors' negative result may simply reflect the differences between their in vitro system and replication machinery in vivo. Those may include the stoichiometry of the fork components, nucleotide pool composition, ionic conditions, etc.

We agree that we cannot exclude the possibility that our reaction conditions are not conducive for triplex formation which may explain why we do not observe significant stalling with (GAA)_n repeats.

Nonetheless, we revisited this by performing the experiments shown in Figure 1B and 1C many more times. In some cases, we could detect very weak but reproducible stalling with the (GAA)₁₆₁ template. This leading strand stalling was considerably weaker than the stall induced by other repeats such as (CGG)₁₆₁ and (CG)₂₄, making it very difficult to study further. Interestingly, stalling induced by (GAA)₁₆₁ was not affected by the presence of pol δ.

We have now replaced panels B and C in Figure 1 with representative experiments and have amended the text in page 8 to reflect this change:

“Replication of substrates containing (CTG)₁₆₁ did not differ from the empty vector control (Fig. 1B, lanes 1-2). However, a very faint 3kb stall band was reproducibly detected with (GAA)₁₆₁ and (CGG)₁₆₁ (Fig. 1B, lanes 3,4). The intensity of this stall was increased when reactions were performed without pol δ for (CGG)₁₆₁ but not for (GAA)₁₆₁ (Fig. 1C). This suggests a role for pol δ in preventing or rescuing leading strand stalls induced by (CGG)₁₆₁. Since these experiments lack components from other pathways, we conclude that the DNA

template itself can induce fork stalling, and that this is modulated by polymerase usage.

Stalling threshold is 17 (CGG)_n repeats and is orientation-dependent

Although both (GAA)₁₆₁ and (CGG)₁₆₁ produced leading strand stalls, further analysis of (GAA)₁₆₁ stalls proved difficult due to the weak signal. We therefore focused on (CGG)_n-induced stalls, which were sufficiently robust when pol δ was absent.”

In addition, we have added the following to our discussion section on GAA repeats:

Finally, we cannot exclude the possibility that our reaction conditions are not conducive for triplex formation.

To sum up, the positive results with GC-rich repeats are impressive, while negative results with other repeats are not convincing at best.

We thank the reviewer for seeing our results as impressive and have now further explored stalling by other repeats such as GAA repeats.

Additional comments:

(1) Lines 180-182 mention that the stall is saturated at CGG61 referencing to Figure 1D. At least by looking at the figure, it seems like CGG81 and CGG161 cause stronger stalls than CGG61. Thus, quantification is a must.

In some of these experiments, the overall efficiency of the reaction can vary, making it difficult to visually compare the relative stall frequency between lanes. As the reviewer requested, we have now added a quantification of five independent experiments, where we normalise the signal of the 3kb stall product to that of the 1.5kb leftward-moving leading strand (run-off) within each lane. This eliminates variation between samples. The result is shown here and is now added as Figure S2A, which reveals that stalling saturates with 41 CGG repeats. The non-zero values seen for no repeats (as well as 11 and 14) is simply due to the signal produced by the gel background, and therefore represents the baseline.

Ratio of stall to run-off signal

Mean of 5 experiments.

Error bars: Standard deviation.

(2) The fact that a lot of experiments in the supplement are done in the absence of Pol δ begs a question: could Pol δ contribute to fork stalling as a whole?

We agree with the reviewer that Pol δ affects fork stalling. Indeed, our results reveal that Pol δ can rescue leading strand stalls that are induced by (CGG)_n and (CG)_n repeats (Figure 3). However, this means that experiments carried out in the presence of Pol δ produce very weak stall signals which are not informative when trying to define sequence requirements. Therefore, the experiments in Figures 1 and 2 (and associated supplements), which define the sequence determinants for stalling by (CGG)_n and (CG)_n, were performed in the absence of Pol δ . However, in all remaining experiments, we have performed assays both in the absence or presence of Pol δ . Since many of these experiments produced identical results regardless of the presence of Pol δ , and for the sake of clarity and completeness, we provided results in the absence of Pol δ in the supplement.

(3) Only repeats forming the most stable alternative DNA structures cause fork stalling in this study. Could it be that in the author's conditions disfavor formation of other secondary structures?

We agree that our assay conditions could either promote or disfavor the formation of different classes of DNA secondary structures. We note that our assay conditions were optimised for efficient replication and were not biased in any way for one type of structure or another. Furthermore, we do not preincubate our DNA substrates to promote structure formation.

The lack of observed stalls for certain sequences could be interpreted in many ways. One possibility is that weak structures do form but cannot inhibit DNA synthesis. Another option is that the folding kinetics of stable structures is faster, allowing them to form within ssDNA that is temporarily exposed between the CMG helicase and Pol ϵ . Alternatively, our assay conditions may not be conducive to certain structures. While we cannot discriminate between these scenarios, we do note that our positive results suggest that our assay conditions can support the formation of three different types of DNA secondary structures (hairpin, G4, iMotif), despite each having very distinct requirements for folding.

(4) Figure 5A: P301R rescuing the stall is clear, but I am not so sure about the Exo- condition. The background in lanes 5-6 and especially 11-12 seems much lower, as well as the amount leading strands. Quantification of these data is needed.

We agree with the reviewer that it is difficult to interpret this result due to the high background. We have therefore performed additional experiments and provide a quantification of five independent assays (quantified in the same fashion as described above for the (CGG)_n series). We have replaced Figure 5A with a more representative result and added within the same panel the quantification shown here.

The results confirm that both mutant forms of pol ε can rescue stalling induced by (CGG)₆₁ and (CG)₂₄. The quantification further reveals that while both mutants rescue (CGG)₆₁ stalls equally well, P301R promotes more efficient rescue of (CG)₂₄ stalls than Exo-.

We have amended the main text on page 14 to reflect these changes:

Leading strand stalls induced by (CGG)₆₁ were significantly weaker in reactions carried out with either pol ε mutants relative to WT (Fig. 5A, lanes 1-6). Stalling produced by (CG)₂₄ was also rescued, but here P301R performed better than Exo- (Fig. 5A, lanes 7-12, also see quantification of five independent experiments).

(5) Lines 344-345 and Fig. 5B lane 14: It looks like addition of dA per se rescues the stall at CGG61 in the presence of Pol δ. This fact is not mentioned or discussed in the MS.

In these experiments we performed a pulse and then a chase. In the pulse we added radiolabelled dATP, along with a low concentration of unlabelled dATP. After 10 min, we added excess unlabelled dATP as well as Pol δ. The rescue seen in lane 14 in Figure 5B is a consequence of adding pol δ rather than excess dATP, as adding dATP alone without Pol δ (see lanes 4/5 and 7/8) did not lead to any rescue.

Mean of 5 experiments.

Error bars: Standard deviation.

(6) Section starting at 353: Pif1 promotes replication through G4. Elsewhere in the paper, the authors suggest that the CGG repeat is forming a hairpin rather than G4-DNA. Yet addition of Pif1 rescues the stall caused by the CGG repeat (Fig. 6A and 6B). Could this observation indicate that CGG are forming G4 as well? Interestingly, knocking down Pif1 in yeast did not cause an increase in fork stalling of CGG repeats in (Anand et al., 2012).

The fact that Pif1 rescues stall induced by (CGG)_n could either mean that Pif1 can unwind hairpins, or that (CGG)_n repeats form a G4. At this point we do not have sufficient evidence to determine the nature of structures formed by (CGG)_n repeats. The literature on this suggests that G4 formation by (CGG)_n repeats is extremely unlikely under physiological conditions and with linear substrates. Rather, hairpins are the predominant product (see for example PMID:14718550). Since our assays utilise linear substrates under physiological pH and salt, we find it highly unlikely that (CGG)_n repeats would form G4 structures. While Pif1 has been studied intensely as a G4 unwinding helicase, there is no evidence that it cannot unwind other structures such as hairpins and iMotifs. Finally, (CG)_n and (CGG)_n produced almost identical results in all our assays. Given that (CG)_n repeats form hairpins, this strongly argues that (CGG)_n repeats also form hairpins.

(7) Lines 358-361: RPA-binding also counteracts formations of other structures, as it was shown to be able to melt both intra- and intermolecular triplexes at concentrations of around 25 nM (Wu et al., 2008). The effect of RPA levels was not, in fact, studied for "all tested sequences" as stated in line 359: it was not done for GAA and CTG repeats (Figure S7 B and C).

We have now performed RPA titrations for these additional substrates (GAA, TTC, CAG and CTG) and find no effect for RPA across a wide range of concentrations (10-200 nM) both in the absence or presence of Pol δ . We have added a new supplemental figure (Fig. S8) with this result.

Reviewer #2 (Remarks to the Author):

The article from Coster and colleges: "The mechanism of replication stalling and recovery within repetitive DNA" uses a innovative research strategy to address the highly significant research question. They use an invitro system that reconstitute DNA replication and monitor changes of replication progression at mono-, di-, and tri-nucleotide repeats. They observed that hairpins as well as quadruplexes cause replication stalling if the structure is located on the leading strand template for DNA replication. They observed that the kind of stall as well as the recovery differs between kind of folded structures (hairpins vs. quadruplexes).

The manuscript is very well and clear written. In particular the introduction, which is very lengthy, is very clear and explains the working hypothesis and the relevance of their story clear. Although the manuscript is very good, I have a few major points that I have listed below:

1. It seems that in the replication assay (e.g. Fig 1) always the right strand is synthesized better (stronger) in comparison to the left? The left strand gets stronger in the pol delta strain (Fig1C). is this due to specific structure formation or strand preference? Why does this biased of strand disappear after the addition of CPD (Fig 1D)?

Detection of replicated products generated by the in vitro replication assay relies on adding a small amount of radiolabelled nucleotide. Since longer products will incorporate more of the labelled nucleotide, the signal for each product is proportional to its size. Therefore, the leading strand product of the rightward moving fork, which is 8.2 kb, is expected to give a signal which is roughly 5-fold that of the leftward moving product, which is 1.5 kb.

The overall efficiency of replication can be variable between experiments, as can be seen with Fig. 1B and 1C. It is therefore difficult to compare efficiency between experiments.

The presence of a CPD in the template leads to stalling of the rightward moving fork, and instead of an 8.2 kb product, we observe a shorter 3 kb product (for more details see [PMID: 29944888](https://pubmed.ncbi.nlm.nih.gov/29944888/)).

2. What portion of the template is folded into the secondary structure?

The reviewer raises an excellent question. To answer this, one would require sensitive and specific methods for detecting different structures. Unfortunately, existing reagents or approaches for secondary structure detection are not sensitive or

specific enough. We have contacted many experts that work on analysis and detection of G-quadruplexes and iMotifs (e.g. David Monchaud, Marie Teulade-Fichoud, Jean Louis Mergny, Marco Di Antonio, Ramon Villar, Zoe Waller). The expert opinion is that detecting a single G4/Motif structure within the context of a dsDNA template is a significant challenge and that current G4 detection reagents are not specific and sensitive enough. Indeed, we have tried a wide range of reagents, including NMM, the BG4 antibody and biotinylated Phen-DC3. While these worked very well with short oligonucleotides, they gave very high non-specific signals with plasmid DNA.

Although we cannot directly detect these structures, we can give a lower estimate of structure formation based on the proportion of stalled forks. For (CG)_n repeats, almost all the forks seem to stall (see Figure 2A). If stalling is indeed due to hairpin formation, they form in almost every instance. In contrast, quadruplex-forming sequences produce a partial stall (Figure 2D and 2E). We estimate that up to 50% of forks stalls, and by extension, that up to 50% of replicated products may generate G4s or iMotifs.

3. In addition it has to be demonstrated that indeed a hairpin or a G4 form. CGG also can fold into G4, less stable than G repeats but indeed a G4. Is the effect really due to hairpin formation or does it also differ in dependency of the stability of the G4. I would like to see other G4 sequences (GG, GGG and GGGG harboring G4). In addition Shape, CD, melting, and other experiments are required to show robust formation of the structure and to estimate stability and conformation status.

As mentioned earlier, there are no methods or reagents available to detect a single G4 in the context of a dsDNA substrate. The point that CGG repeats can also form G4 structures was also raised by reviewer #1. The literature on this suggests that G4 formation by (CGG)_n repeats is extremely unlikely under physiological conditions and with linear substrates. Rather, hairpins are the predominant product (see for example [PMID:14718550](https://pubmed.ncbi.nlm.nih.gov/14718550/)). Since our assays utilise linear substrates under physiological pH and salt, we find it highly unlikely that (CGG)_n repeats would form G4 structures. Also, the fact that stalling by both (CG)_n and (CGG)_n repeats was rescued by Pol δ, but stalling by (G)_n was not, suggests that both form a similar structure which is distinct from that formed by (G)_n.

Indeed, an important question is whether stalling by (G)_n is unique to this homopolymer, or whether other G4-forming sequences can induce stalling. As the reviewer points out, it is important to test a range of G4-forming sequences with defined stabilities and characterise them using biophysical methods. This question has formed the basis of an entire separate project in the lab which we think is

beyond the scope of this manuscript and will be submitted for publication in the near future as an extension of this study.

4. Are all assays been performed under physiological salt conditions? Are the reaction time, pausing differ in response to salt changes? it has been known for example that salt concentration and the type of salt alters the conformation of G4 and other structures. E.g. Li reduces G4 potential whereas K⁺ enhances and stabilizes G4s

Our standard assay conditions are performed under mild salt concentration and physiological pH (pH=7.6, 200 mM KGlutamate). However, our assay requires the addition of a large number of purified proteins which are each soluble in their own unique buffer. These buffers contribute a certain amount of ions, including Na⁺ and K⁺, making it difficult to fully control the presence of one specific ion (as is possible for example with isolated oligos).

Nonetheless, we have tried changing the main salt to Na⁺ or Li⁺, but see no difference in stalling induced by (G)_n. Given the complex ionic composition of our reactions, it is difficult to draw from this result any concrete conclusion.

5. They demonstrated beautifully that CTG and GAA repeats do not stall replication, only if replication fork fidelity is reduced (dNTP levels, Pol delta..) can they discuss changes observed in repeat expansion patients to intrinsic changes?

While our work follows the dynamics of replication, we cannot at this point comment on replication accuracy. In other words, our results lack the resolution to tell if repeats have expanded or contracted. In addition, numerous genetic studies show that repeat expansion occurs orders of magnitude less frequently than stalling, often requiring clever genetic reporters to allow for their selection and detection over multiple cell doublings (e.g. [PMID: 12556494](https://pubmed.ncbi.nlm.nih.gov/12556494/)). We therefore do not expect to detect significant repeat expansion or contraction within a single round of replication.

6. They elucidated that Pol delta overtime unfold hairpins at repeats. Can they be more specific? What is the timing? Is this relevant in vivo?

To provide a more accurate answer to this question we performed a pulse-chase time course with more discrete time points. This reveals that Pol δ can resolve stalls induced by hairpin-forming repeats within minutes. Our earliest time point shows that the stall is practically gone for both (CGG)₆₁ and (CG)₂₄ after 2.5 min of adding pol δ . We have now replaced Figure 3F with the time-course of (CG)₂₄ and added the (CGG)₆₁ time-course as Figure S5A.

We have amended the main text on page 12 to reflect these changes:

Figure 3F shows that a clear stall with (CG)₂₄ was evident after the 10 min pulse (lane 1), which remained unaltered in the absence of pol δ for 30 min (lanes 2-6). However, pol δ was able to resolve this stall within 2.5 min (Fig. 3F, lane 7) and similar rescue kinetics were also observed with (CGG)₆₁ (Fig. S5A).

7. They observed that PCNA assist directly in replication progression. Is this a direct role of PCNA that participate in structure resolution?

Our result in Fig. S5 shows that PCNA is required for recovery from stalls, but this is only true when Pol δ is present. In the absence of Pol δ , adding PCNA had no effect. This suggests that the major role of PCNA in these reactions is enhancing the processivity of Pol δ . To clarify this point we added the following sentence when describing this result in page 12:

Importantly, adding PCNA in the absence of pol δ had no effect.

8. Can they exclude that G4 on the lagging strand will cause replication pausing? Or is the system not specific for lagging strand changes? it would be nice to discuss literature on structure on the lagging strand in more detail and discuss the power and the draw back of the method more clearly.

In a separate project in the lab, we have compared different G4-forming motifs and found that when the G4-forming sequence is on the lagging strand template, leading strand synthesis is not affected. This suggests that even if a G4 forms on the lagging strand, it does not affect CMG and Pol ϵ . Indeed, it has been shown in vitro that a DNA lesion on the lagging strand that completely prevents synthesis does not affect leading strand progression (PMID: 29944888).

Detecting lagging strand stalls in our system is far more challenging than leading strand stalls because of its inherent discontinuous nature. Even if the replisome encounters problems on the lagging strand, it can easily reprime beyond the lesion or structure, leaving behind a short gap. This is further complicated by the fact that most of our substrates contain a stalling sequence on both strands. For example, (CG)₂₄ would have the same sequence on both leading and lagging strand templates.

Despite these challenges, we have tried to assess the presence of lagging strand gaps by inducing lagging strand maturation by adding the flap endonuclease Fen1 and the ligase Cdc9. A caveat of this approach is that we cannot assess the role of Pol δ because it is essential for lagging strand maturation.

As expected, in the experiment shown here, no stalls were seen with the two hairpin-forming repeats, as pol δ can resolve them. Interestingly, we consistently observe a novel 5 kb band with both (G)₅₀ and (C)₅₀, which is the expected size of a product formed when the entire lagging strand is fully ligated except for a single gap positioned within the repeats. This band is significantly weaker than the leading strand stalls, suggesting that if lagging strand structures do form and stall synthesis, they do so at a lower frequency.

Another drawback of this approach is that the 5 kb band could also arise by a repriming event on the leading strand. Taking all these considerations into account, we are reluctant to make any strong statements regarding lagging strand stalls or structure formation. While this question is important and merits further future work, we feel it is beyond the scope of this manuscript.

9. Can other helicases, in addition to Pif1 (DNA2 is not an unwinding helicase in vitro) also support replication progression: e.g. BLM, WRN, FANCI?

The most relevant helicases to test in this context are budding yeast helicases that have been implicated in countering repeat instability – Srs2 and Sgs1. We have obtained published expression vectors and followed established protocols and made many attempts at expressing and purifying both of these helicases. Although we were able to generate proteins of sufficient amount and purity, these were not convincingly active in standard helicase unwinding assays. In the case of Srs2 we also obtained purified protein from Dr. Tom Deegan (University of Edinburgh). However, although this reagent was published to work in one specific unwinding assay (using a circular ssDNA template and annealed oligo in [PMID: 30850330](https://pubmed.ncbi.nlm.nih.gov/30850330/)), in our hands it was not active using standard annealed oligos as substrates. Importantly, using similar conditions and substrates we could observe robust activity of Pif1 and the CMG helicase. It is therefore unclear why this prep of Srs2 was active with one substrate but not another.

We have also obtained purified human BLM and WRN, both full length as well as helicase domain (HD) only, from the labs of Dr. Antony Oliver and Laurence Pearl (GDSC, University of Sussex). They have demonstrated that these are active using a fluorescence-based assay.

We tested whether these helicases were able to rescue leading strand stalls produced by (CGG)₆₁, (CG)₂₄, (G)₅₀ and (C)₅₀. While Pif1 exhibited robust rescue with all substrates tested, none of these helicases were able to support replication past the repeats, as seen in these representative examples.

We are aware that these experiments are quite artificial in that we are adding human helicases to a budding yeast replisome. It is possible that BLM and WRN may require species-specific interactions and that this may explain the lack of any apparent rescue. Regarding Srs2, we are not convinced that the protein we obtained exhibits robust helicase activity on our model substrates.

Altogether, and despite much effort, we feel that these results are not robust enough for any strong conclusions and we are not convinced that their value outweighs their caveats.

Reviewer #3 (Remarks to the Author):

The authors have systematically studied how repetitive sequences affect leading strand synthesis in a reconstituted in vitro replication system. Earlier studies have demonstrated that both DNA polymerase epsilon and DNA polymerase delta have difficulties bypassing repetitive DNA sequences that can form secondary structures. Here the authors demonstrate that such repetitive sequences impede DNA synthesis, even immediately after the template strand passed through the replicative helicase and should thus be relatively free of secondary structures. Furthermore, the authors suggest that the resulting uncoupling between Pol epsilon and CMG may induce check-point activation similar to when DNA lesions are present in the leading strand. This remains to be shown. Finally the authors demonstrate that Pol delta can assist Pol epsilon in bypassing the secondary structure, and that Pif1 is very efficient in suppressing the stalling by presumably resolving the secondary structure of the DNA. A more succinct discussion would improve the manuscript.

We thank the reviewer for these comments. As suggested, we have shortened the discussion section to make it more succinct.

REVIEWERS' COMMENTS

Reviewer #1 (Remarks to the Author):

The revised manuscript is significantly improved. The authors were able to add new data and interpretations that satisfactorily addressed most of the reviewers' comments.

Most importantly, the authors now detect a weak but reproducible stalling in the (GAA)₁₆₁ template (revised Fig. 1B). This observation would undoubtedly broaden the appeal of this study to the genome instability crowd. Given that, the sentence in the Discussion "Altogether, we conclude that within our experimental conditions, (GAA)_n repeats by themselves do not cause significant leading strand stalling" seems out of context.

Another important improvement is that the authors now acknowledge and discuss that their assay conditions (pH=7.6, 200 mM KGlutamate) could either promote or disfavor the formation of different classes of DNA secondary structures. Notably, intracellular pH in yeast is significantly lower (~6.5) than their pH. Since stability of triplexes and CGG-quadruplexes depends on pH, this may explain, at least partly, the differences between their in vitro and yeast in vivo data. Something to pursue in the future?

Several of the authors' statements regarding previously published data are factually incorrect. Below are just two examples, but there are some more:

(1) "This is in agreement with earlier studies showing that in bacteria, two tracts of (GAA)_n can form triplexes, whereas a single tract cannot(ref 74)"

In fact, ref. 74 characterized the formation of the so-called sticky DNA in bacterial plasmids in vitro, which indeed requires the presence of two distinct GAA runs within the same plasmid. It was unambiguously demonstrated that individual GAA runs in bacterial plasmids form canonical H-DNA (DOI: 10.1093/nar/gkh274).

Overall, I would advise the authors not to go too deeply in discussing sticky DNA, as its structure is far from clear.

(2) "Fiber labelling of individual replication forks in the CGG-expanded FMR1 locus from Fragile X syndrome cells revealed very little stalling (ref 25)"

In fact, the Abstract of ref 25 states explicitly: "Examining DNA replication fork progression on single DNA molecules at the endogenous FMR1 locus revealed that replication forks stall at CGG repeats in human cells".

The authors need to double-check the accuracy of their quotations.

Reviewer #2 (Remarks to the Author):

We congratulate the author on this nice work. They have clearly addressed all my concerns. One super minor comment, maybe they should state in the discussion that more experiments are required to fully understand the role of G4s in replication stalling. but I think for this current publication it is perfect like it is

Reviewer #1 (Remarks to the Author):

The revised manuscript is significantly improved. The authors were able to add new data and interpretations that satisfactorily addressed most of the reviewers' comments.

We thank the reviewer for seeing the improvements in our manuscript.

Most importantly, the authors now detect a weak but reproducible stalling in the (GAA)₁₆₁ template (revised Fig. 1B). This observation would undoubtedly broaden the appeal of this study to the genome instability crowd. Given that, the sentence in the Discussion "Altogether, we conclude that within our experimental conditions, (GAA)_n repeats by themselves do not cause significant leading strand stalling" seems out of context.

We have now modified the discussion so that this sentence now reads:

"Altogether, we conclude that within our experimental conditions, (GAA)_n repeats by themselves cause mild leading strand stalling"

Another important improvement is that the authors now acknowledge and discuss that their assay conditions (pH=7.6, 200 mM KGlutamate) could either promote or disfavor the formation of different classes of DNA secondary structures. Notably, intracellular pH in yeast is significantly lower (~6.5) than their pH. Since stability of triplexes and CGG-quadruplexes depends on pH, this may explain, at least partly, the differences between their in vitro and yeast in vivo data. Something to pursue in the future?

We agree with the reviewer that assay conditions may have a profound effect on structure formation. The challenge for future work is to manipulate conditions in a way that does not inhibit the replication reaction itself.

Several of the authors' statements regarding previously published data are factually incorrect. Below are just two examples, but there are some more:

(1) "This is in agreement with earlier studies showing that in bacteria, two tracts of (GAA)_n can form triplexes, whereas a single tract cannot(ref 74)"

In fact, ref. 74 characterized the formation of the so-called sticky DNA in bacterial plasmids in vitro, which indeed requires the presence of two distinct GAA runs within the same plasmid. It was unambiguously demonstrated that individual GAA runs in bacterial plasmids form canonical H-DNA (DOI: 10.1093/nar/gkh274).

Overall, I would advise the authors not to go too deeply in discussing sticky DNA, as its structure is far from clear.

We thank the reviewer for highlighting these inaccuracies. We have accordingly shortened the section on GAA repeats in the discussion.

(2) "Fiber labelling of individual replication forks in the CGG-expanded FMR1 locus from Fragile X syndrome cells revealed very little stalling (ref 25)"

In fact, the Abstract of ref 25 states explicitly: "Examining DNA replication fork progression on single DNA molecules at the endogenous FMR1 locus revealed that replication forks stall at CGG repeats in human cells".

The authors need to double-check the accuracy of their quotations.

Relative to stalling observed with GAA-expanded cells from Friedreich's Ataxia derived cells (Ref 26), CGG stalls are much milder (Referred to as pauses in Ref 25). We therefore did not state that there was no stalling, but rather little stalling. We have now amended the text to clarify this point, which now reads:

"Fiber labelling of individual replication forks in the CGG-expanded FMR1 locus from Fragile X syndrome cells revealed stalling²⁵."

Reviewer #2 (Remarks to the Author):

We congratulate the author on this nice work. They have clearly addressed all my concerns. One super minor comment, maybe they should state in the discussion that more experiments are required to fully understand the role of G4s in replication stalling. but I think for this current publication it is perfect like it is

We thank the reviewer for the kind words. We have added the following text in the discussion to highlight the importance of further studying G4 replication:

It is worth noting that we have only tested a single G4-forming sequence and a single i-motif forming sequence. These homopolymers may not accurately represent how other quadruplex-forming sequences behave. Therefore, an important area of future study is to establish how other quadruplex-forming sequences affect replication.